# Towards Universal Gene Regulatory Network Inference: Unlocking Generalizable Regulatory Knowledge in Single-cell Foundation Models

Jiaxin Qi [1 2]   Hang Li [2]   Yan Cui [2]   Yuhua Zheng [2]   Jianqiang Huang[† 1 2 3]

## Abstract

Gene Regulatory Network (GRN) inference is essential for understanding complex cellular mechanisms, rendered tractable through single-cell transcriptomic data. With the emergence of single-cell Foundation Models (scFMs), enhanced transcriptomic encoding is widely expected to revolutionize GRN inference. However, we observe that their performance remains far from satisfactory. The primary reason is that the standard reconstruction-based pre-training objectives often fail to explicitly capture latent regulatory signals. To bridge this gap, we first introduce a GRN generalization benchmark designed to evaluate regulatory predictions on unseen genes and datasets, which relies on the zero-shot capabilities of scFMs and is inherently challenging for traditional methods. Furthermore, to probe whether scFMs encode generalizable inter-gene representations for GRN inference, we propose two novel methods, Virtual Value Perturbation and Gradient Trajectory, to distill implicit regulatory information from scFMs into highly generalizable inter-gene features. Extensive experiments demonstrate that our approach significantly outperforms existing methods, establishing a new paradigm for leveraging the potential of scFMs in universal GRN inference.

## 1. Introduction

Gene Regulatory Networks (GRN) constitute the core mechanisms that govern complex biological processes by encoding the intricate causal dependencies among genes (David-

[1]Computer Network Information Center, Chinese Academy of Sciences, Beijing, China [2]Hangzhou Institute for Advanced Study, University of Chinese Academy of Sciences, Hangzhou, China [3]University of Chinese Academy of Sciences, Beijing, China. Correspondence to: Jianqiang Huang <jqhuang@cnic.cn>.

*Proceedings of the 43rd International Conference on Machine Learning*, Seoul, South Korea. PMLR 306, 2026. Copyright 2026 by the author(s).

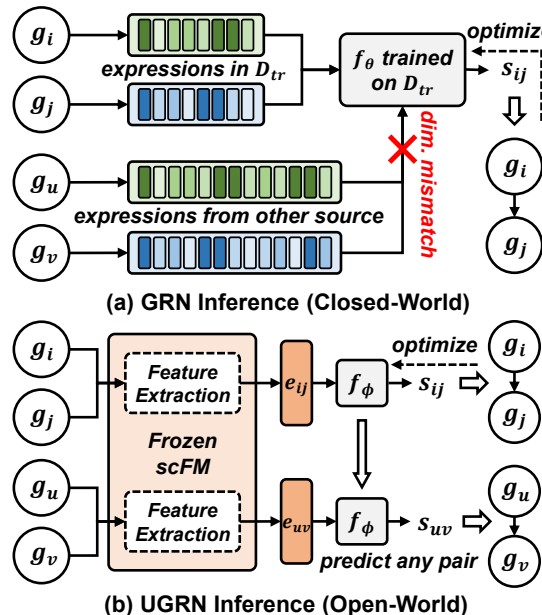

**(a) GRN Inference (Closed-World)**

**(b) UGRN Inference (Open-World)**

*Figure 1.* (a) Traditional GRN inference operates in a closed-world setting, where optimized $f_\theta$ struggles with dimension mismatches on unseen genes from heterogeneous datasets. (b) Our UGRN setting utilizes frozen scFMs for universal feature extraction, enabling the generalization of regulatory predictions by "translator" $f_\phi$ to open-world scenarios involving unseen genes and datasets.

son et al., 2002). Recent advancements in single-cell RNA sequencing (scRNA-seq) have provided high-resolution, cell-level gene expression profiles, enabling the estimation of GRNs directly from observational data (Aibar et al., 2017; Pratapa et al., 2020). For example, if the expression of gene $g_i$ is consistently correlated with that of $g_j$, a potential regulatory link is hypothesized (Eisen et al., 1998). Traditionally, as shown in Figure 1(a), GRN inference has relied on identifying co-expression dependencies between genes within specific datasets (Song et al., 2012). However, these methods are often constrained to a "closed-world" setting, where they remain effective only for genes observed during training and struggle to generalize to unseen genes from heterogeneous datasets. This limitation primarily stems from the lack of unified expression manifolds and generalizable regulatory representations between genes.

Recently, Single-Cell Foundation Models (scFMs) (Hao et al., 2024; Yang et al., 2024) have emerged as a promising paradigm for addressing various downstream biological tasks, including GRN inference. By pre-training on large-scale gene expression data via self-supervised objectives, e.g., masked value reconstruction, scFMs are expected to capture profound biological priors for inter-gene relationships. Consequently, an increasing number of studies attempt to leverage scFMs for zero-shot GRN inference through two primary strategies: (1) In-silico Perturbation, which simulates biological knockouts by zeroing out the input expression of a source gene $g_i$ to observe the response of a target gene $g_j$ (Theodoris et al., 2023; Cui et al., 2024); and (2) Attention-based analysis, which interprets the model's attention weights as proxies for regulatory strength to predict inter-gene relations (Yang et al., 2022; Cui et al., 2024). Despite their theoretical appeal, recent works point out that these methods often exhibit suboptimal performance, sometimes even failing to surpass random prediction (Jin et al., 2025; Ahlmann-Eltze et al., 2025). This phenomenon may stem from the misalignment between the reconstruction-based pre-training and the downstream requirements of inter-gene inference, leading to growing skepticism regarding the empirical efficacy of scFMs within the biological community (Wu et al., 2025).

In this paper, we argue that scFMs possess rich and transferable regulatory knowledge, but current methods are too simplistic to bridge the gap between correlation-based reconstruction output and gene regulation. For example, the influence of gene $g_i$ on $g_j$ estimated via simple "zero-out" perturbations merely reflects the model's reliance on $g_i$ to reconstruct $g_j$, which represents neither true causality nor the full correlation between them. However, we posit that such conditioned inter-gene influences reflect a consistent internal understanding of gene relationships within the scFMs. As shown in Figure 1, the challenge lies in "translating" this latent knowledge into gene regulatory insights. To this end, we introduce a novel setting termed Universal Gene Regulatory Network inference. This paradigm requires the model to learn a mapping (a "translator") from scFM-derived inter-gene knowledge to ground-truth GRNs using source genes and then generalize to unseen genes from heterogeneous datasets. This setting serves as a rigorous benchmark for evaluating whether foundation models can capture generalizable regulatory principles, a capability unattainable by traditional statistical methods constrained to dataset-specific expression understanding.

We first establish two baselines by utilizing gene embeddings and traditional "zero-out" perturbation influences between genes as features to map onto gene regulatory relationships, thereby constructing a benchmark for our universal GRN setting. To derive more profound and generalizable inter-gene features from scFMs, we propose two novel

methods, Virtual Value Perturbation (VVP) and Gradient Trajectory (GDT). For VVP, observing that distinct genes possess different baseline expressions, we point out that the "zero-out" operation introduces inconsistent perturbation magnitudes. Consequently, we standardize this process by employing a unified virtual value as the base expression, ensuring that gene interactions are queried under a consistent reference value. Furthermore, since scFMs can encode and reconstruct arbitrary expression values beyond the observed cellular range, we define a series of perturbation target values, rather than a single zero, to extract richer inter-gene influences. For GDT, as discrete perturbations only represent the influence over a change interval, we argue that the influence at specific expression levels should be characterized by gradients. Leveraging the gradient backpropagation of scFMs, we propose the Gradient Trajectory method, which extracts gradients along a series of virtual values, to reflect the gene relationship across varying expression levels.

Extensive experiments across multiple datasets and different settings demonstrate that our proposed methods significantly outperform traditional scFM-based methods and our baselines. The results show that when properly queried, scFMs exhibit a stable and profound understanding of GRNs that far exceeds random expectation, thereby validating the utility of large-scale pre-training for single-cell transcriptomics. Notably, because our method utilizes virtual values to extract inter-gene features for GRN inference, it can predict regulatory links even in the absence of real-world expression measurements, providing a powerful tool for constructing Universal Gene Regulatory Networks.

Our main contributions are summarized as follows:

- We analyze scFM-based GRN inference, identifying the critical misalignment between reconstruction objectives and gene regulation that limits current methods.

- We introduce the UGRN framework, a benchmark probing whether scFMs encode generalizable inter-gene representations for GRN inference across unseen genes and heterogeneous datasets.

- We propose VVP and GDT, achieving SOTA performance in extensive experiments and ablations by extracting generalizable regulatory features from scFMs.

## 2. Related Works

**Gene Regulatory Network Inference**. Inferring GRNs from single-cell transcriptomics has evolved from statistical approaches to deep learning frameworks. Traditional methods utilize tree-based regression (Huynh-Thu et al., 2010; Moerman et al., 2019) or information-theoretic metrics (Margolin et al., 2006; Faith et al., 2007) to capture non-linear co-expression patterns. More recent deep learn-

ing approaches, such as GNNs and VAEs, explicitly model network topology to learn latent representations of gene dependencies (Chen & Liu, 2022; Shu et al., 2021). However, these methods predominantly operate under a "closed-world" assumption, where they capture the manifold of specific datasets and cannot generalize to unseen genes without re-training (Kedzierska et al., 2025). In contrast, we introduce a Universal GRN setting. Instead of fitting dataset-specific distributions, we leverage pre-trained foundation models to extract generalizable regulatory features, enabling generalization to unseen genes and heterogeneous datasets.

**Single-cell Foundation Models**. Inspired by Large Language Models, Single-cell Foundation Models (scFMs) such as scGPT (Cui et al., 2024), Geneformer (Theodoris et al., 2023), and scBERT (Yang et al., 2022) learn transcriptomic representations via masked value modeling on massive cell atlases. Current strategies for GRN inference with scFMs rely on simple heuristics, either by interpreting raw attention weights as regulatory strength (Kalfon et al., 2025; Yang et al., 2022) or by performing in-silico perturbation by zeroing out input genes (Theodoris et al., 2023; Cui et al., 2024). However, recent benchmarks reveal that these methods often yield suboptimal performance (Kedzierska et al., 2025; Jin et al., 2025; Ahlmann-Eltze et al., 2025; Wu et al., 2025). We argue that this failure stems from the simplistic implementation of scFMs rather than its lack of regulatory knowledge. Unlike direct heuristic mapping, we propose Virtual Value Perturbation and Gradient Trajectory to actively distill implicit regulatory signals from frozen scFMs, bridging the gap between correlation-based features and gene regulations.

## 3. Method

### 3.1. Preliminaries

Current GRN inference research mainly characterizes regulatory dependencies between individual gene pairs. Following (Wang et al., 2024), we treat GRN inference as a pair-wise prediction task. Note that, although the global GRN topology constitutes essential background knowledge (Barabási & Oltvai, 2004), such information is practically incorporated into the latent gene embeddings to facilitate pair-wise learning objectives. Thus, for simplicity, we omit the explicit global network formulation and focus on pair-wise learning.

**GRN Inference**. Given a set of genes $\mathcal{G} = \{g_1, \ldots, g_K\}$ and a single-cell gene expression matrix $\mathbf{X} \in \mathbb{R}^{N \times K}$, let $\mathbf{x}_i \in \mathbb{R}^N$ denote the expression vector of gene $g_i$ across $N$ observed cells (corresponding to the $i$-th column of $\mathbf{X}$). We have access to ground-truth regulatory annotations, formalized as a training set $\mathcal{D}_{tr} = \{(\mathbf{x}_i, \mathbf{x}_j, y_{ij})\}_{(i,j) \in \Omega_{tr}}$, where $y_{ij} \in \{0, 1\}$ indicates whether $g_i$ regulates $g_j$, and $\Omega_{tr}$ de-

notes the set of observed pair indices.

The fundamental objective of GRN inference is to learn a parameterized mapping function $f_\theta$ from expression to regulatory probabilities based on $\mathcal{D}_{tr}$, to generalize to unseen pairs in a held-out test set $\mathcal{D}_{te}$ under the same expression matrix $\mathbf{X}$. Formally, for any pair $(g_i, g_j)$ in $\mathcal{D}_{te}$, the model predicts a regulatory score

$$s_{ij} = f_\theta(\mathbf{x}_i, \mathbf{x}_j), \ (i, j) \in \Omega_{te}. \tag{1}$$

**Traditional Methods**. Existing methods typically employ linear regression or deep neural networks to learn the mapping $f_\theta$ by minimizing the empirical risk over $\mathcal{D}_{tr}$ (Haury et al., 2012; Yuan & Bar-Joseph, 2019). The training objective is formulated as the binary cross-entropy loss:

$$\mathcal{L}_\theta = - \sum_{(i,j) \in \Omega_{tr}} [y_{ij} \log s_{ij} + (1 - y_{ij}) \log(1 - s_{ij})], \tag{2}$$

where $s_{ij}$ denotes the predicted regulatory probability between gene $i$ and gene $j$ defined in Eq. (1).

However, this paradigm relies on the assumption that the training and testing data share the same expression distribution and graph structure. Since the dimension of features $\mathbf{x}_i$ is tied to the number of cells in $\mathcal{D}_{tr}$, the learned function $f_\theta$ is bound to this fixed size. Consequently, when applied to a new dataset with a distinct cell count $N'$, the misaligned expression dimension renders the model ineffective. Furthermore, if the method explicitly encodes the global topology of $\mathcal{G}$ (Chen & Liu, 2022), it becomes unable to generalize to novel gene sets $\mathcal{G}'$ with unseen network topologies.

**scFM-based Methods.** Single-Cell Foundation Models (scFMs) (Cui et al., 2024) leverage large-scale pre-training on massive single-cell corpora to encode gene representations and the inter-gene interaction mechanism. These models encode a unified gene vocabulary $\mathcal{V}$ (where $|\mathcal{V}| \gg K$), capturing complex regulatory semantics that facilitate *zero-shot* GRN inference without explicit supervision from $\mathcal{D}_{tr}$.

Formally, let $\mathcal{M}$ denote a Transformer-based scFM. The model takes a cell's expression vector $\mathbf{x}_c \in \mathbb{R}^K$ (corresponding to a row in $\mathbf{X}$) as input and outputs a reconstructed vector $\hat{\mathbf{x}}_c = \mathcal{M}(\mathbf{x}_c)$ in the same feature space. Leveraging this input-output architecture, existing approaches primarily employ two strategies to estimate the regulatory score $s_{ij}$.

*(1) In-Silico Perturbation*. This approach simulates biological knockout experiments. It measures the regulatory effect of gene $g_i$ on $g_j$ by computing the shift in the predicted expression of $g_j$ when the input value for $g_i$ is set to zero. For a given cell expression $\mathbf{x}_c$, the regulatory score is defined as

$$s_{ij} = \mathcal{M}(\mathbf{x}_c)_j - \mathcal{M}(\mathbf{x}_{c \neg i})_j, \tag{3}$$

where $\mathcal{M}(\cdot)_j$ represents the reconstructed expression for gene $j$, $\bar{\mathbf{x}}_{c \neg i}$ denotes the cell expression for $c$ with gene $i$

set to zero. In practice, $s_{ij}$ is computed either by using the dataset-level mean expression as $\mathbf{x}_c$, or by averaging the perturbation scores across all individual cells in $\mathcal{D}_{tr}$.

*(2) Attention Extraction.* This approach hypothesizes that the internal attention weights of Transformers implicitly encode regulatory dependencies. By forwarding a cell expression $\mathbf{x}_c$ through $\mathcal{M}$, the regulatory score for $g_i$ to $g_j$ is defined as

$$s_{ij} = \sum_l \text{Att}^{(l)}(\mathcal{M}, \mathbf{x}_c)_{ij}, \qquad (4)$$

where $\text{Att}^{(l)}(\mathcal{M}, \mathbf{x}_c)_{ij}$ denotes the attention weight between gene $i$ and gene $j$ in the $l$-th layer, extracted from the forward pass of $\mathbf{x}_c$ through $\mathcal{M}$, and the summation aggregates these weights across all layers.

Despite their theoretical appeal for zero-shot GRN inference, these methods significantly underperform supervised baselines (Wang et al., 2024; Kommu et al., 2025). We argue that such straightforward heuristics, Eqs. (3) and (4), fail to explicitly represent regulatory effects, as they merely reflect the model's gene reconstruction capability. Furthermore, the standard GRN evaluation setting is hard to assess the generalization power of scFMs. To address these gaps, we propose a novel framework designed to unlock the true potential of scFMs for gene regulatory inference.

### 3.2. Universal Gene Regulatory Network Inference

To fully unlock the potential of scFMs and bridge the gap between model predictions and regulatory relationships, we propose a novel framework termed Universal Gene Regulatory Network (UGRN) Inference. This framework learns a "translator" mapping scFM-derived inter-gene features to regulatory probabilities. By exploiting the unified vocabulary and transferable knowledge of scFMs, UGRN facilitates regulatory prediction for any arbitrary gene pairs.

**Problem Definition.** Let $\mathcal{M}$ denote an scFM, and we assume access to a base GRN dataset $\mathcal{D}_b = \{\mathbf{X}_b, \Omega_b\}$ defined over a gene set $\mathcal{G}_b$. The goal of UGRN is to learn a projector $f_\phi$, which takes pair-wise features $\mathbf{e}_{ij}$ extracted by $\mathcal{M}$ for $g_i$ and $g_j$ and predicts their regulatory link

$$s_{ij} = f_\phi(\mathbf{e}_{ij}), \ g_i, g_j \notin \mathcal{G}_b, \qquad (5)$$

where $f_\phi$ can be trained by Eq. (2). During inference, both the expression $\mathbf{X}$ and the gene set $\mathcal{G}$ differ from those in the training. Therefore, the core challenge of UGRN is how to extract a more generalizable $\mathbf{e}_{ij}$ from scFMs.

**Baseline Methods.** To establish benchmarks for our setting, we propose two straightforward strategies to derive $e_{ij}$.

*(1) Perturbation-based Features.* Instead of using the raw value change from In-Silico Perturbation directly as the regulatory prediction, we treat the perturbation response as

an inter-gene feature. Based on the formulation in Eq. (3), we derive the feature $e_{ij}$ as

$$e_{ij} = \mathcal{M}(\bar{\mathbf{x}})_j - \mathcal{M}(\bar{\mathbf{x}}_{\neg i})_j, \qquad (6)$$

where $\bar{\mathbf{x}}$ denotes the mean expression averaged across all cells in the given dataset, with other notations remaining consistent with Eq. (3). This strategy effectively standardizes the input dimension, thereby mitigating discrepancies caused by varying cell counts across different GRN datasets.

*(2) Embedding-based Features.* Since scFMs are pre-trained to reconstruct expressions across massive gene corpora, the learned gene vocabulary embedding $\mathbf{E}_\mathcal{M} \in \mathbb{R}^{|\mathcal{V}| \times d}$ naturally serves as an intrinsic encoding for genes, where $|\mathcal{V}|$ denotes the vocabulary size and $d$ is the hidden dimension. For a gene pair $(g_i, g_j)$, the pair-wise feature can be directly constructed by summing their embeddings

$$\mathbf{e}_{ij} = \mathbf{E}_{\mathcal{M},i} + \mathbf{E}_{\mathcal{M},j}, \qquad (7)$$

where $\mathbf{E}_{\mathcal{M},i} \in \mathbb{R}^d$ denotes the gene embedding of $g_i$.

While these baselines validate the feasibility of the UGRN setting, they still represent a simplistic exploitation of the foundation models. To fully unlock the potential of scFMs, we require methods that can deeply probe the model for comprehensive inter-gene representations.

### 3.3. Our Methods

We propose Virtual Value Perturbation (VVP) and Gradient Trajectory (GDT) to serve as the advanced extractors for $\mathbf{e}_{ij}$. Both are motivated by the need to derive the comprehensive influence of $g_i$ on $g_j$ via counterfactual reasoning.

**Virtual Value Perturbation (VVP).** Conventional perturbation estimates the response of a target gene $g_j$ by zeroing out $g_i$ from its observed value $\mathbf{x}_{c,i}$ under cell $c$. This approach inherently suffers from scale inconsistency, where the perturbation magnitude is coupled with the original value of $g_i$, rendering predicted influences incomparable across different genes from heterogeneous datasets.

To decouple perturbation features from dataset statistics, we utilize the capability of scFMs, which can process arbitrary expression values beyond the observations. Thus, we introduce a unified virtual base value $v_b$ to serve as a consistent reference for all genes. Instead of querying the model with a simple "on/off" state, we perturb the source gene $g_i$ from $v_b$ to another virtual target value $v_p$, and the corresponding inter-gene feature is formulated as:

$$e_{ij}^{v_p} = \mathcal{M}(\mathbf{v}_{g_i \leftarrow v_p})_j - \mathcal{M}(\mathbf{v}_{g_i \leftarrow v_b})_j, \qquad (8)$$

where $\mathbf{v}_{g_i \leftarrow v}$ denotes virtual cell expression with the entry for $g_i$ set to $v$ and other genes are fixed at $v_b$. To capture comprehensive dynamics, we define a set of target values

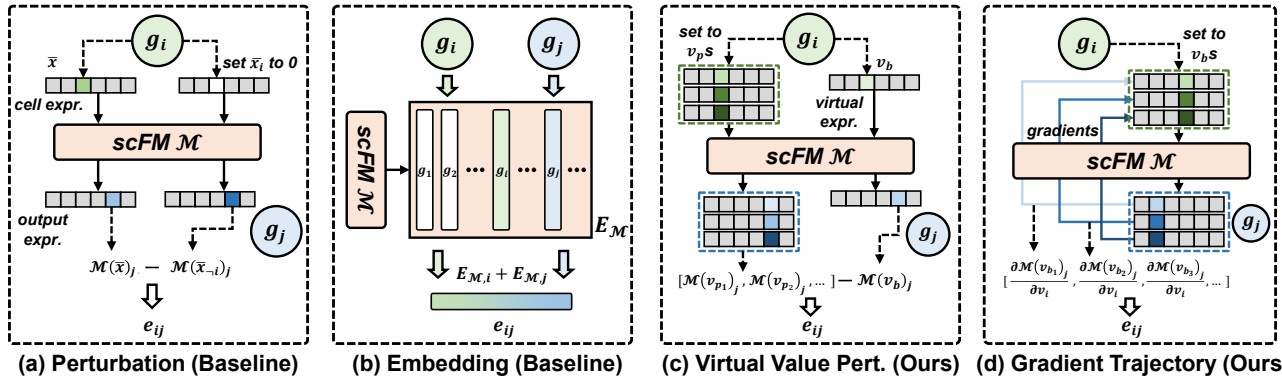

*Figure 2.* Illustration of methods for extracting $\mathbf{e}_{ij}$ from scFM $\mathcal{M}$. The figure compares baselines (a) Perturbation and (b) Embedding against our proposed (c) Virtual Value Perturbation and (d) Gradient Trajectory. Green and blue blocks highlight the entries corresponding to source gene $g_i$ and target gene $g_j$, respectively, while grey blocks represent background genes. Note that (a) utilizes real mean expression $\bar{x}$, whereas (c) and (d) employ virtual expression vectors. Mathematical notations are simplified for visual clarity.

$\{v_{p,1}, \ldots, v_{p,M}\}$ and construct $\mathbf{e}_{ij}$ by concatenating the corresponding responses, $\mathbf{e}_{ij} = [e_{ij}^{v_{p,1}}; \ldots; e_{ij}^{v_{p,M}}]$, which serve as a comparable representation for all gene pairs regardless of their observational data.

**Gradient Trajectory (GDT).** Since VVP essentially represents the regulatory influence induced by expression variations over an interval, we argue that instantaneous regulatory mechanics are characterized by local sensitivity.

Leveraging the differentiability of scFMs, we propose to extract the instantaneous rate of change via gradient backpropagation. We define an ordered set of virtual base values $\{v_{b,1}, \ldots, v_{b,T}\}$ covering the dynamic range. For each state $v_{b,t}$, we compute the partial derivative of the reconstruction of target gene $j$ with respect to the input of source gene $i$, and concatenate them into a trajectory vector

$$\mathbf{e}_{ij} = [\nabla_{ij}^{(1)}; \ldots; \nabla_{ij}^{(T)}]; \ \nabla_{ij}^{(t)} = \frac{\partial \mathcal{M}(\mathbf{v}_{g_i \leftarrow v_{b,t}})_j}{\partial v_i}, \quad (9)$$

where $\mathbf{v}_{g_i \leftarrow v_{b,t}}$ denotes the virtual input vector with the entry for $g_i$ set to $v_{b,t}$, $v_i$ denotes the scalar input corresponding to gene $g_i$ in $\mathbf{v}$, and $[;]$ represents the concatenation operation. By organizing these local sensitivities into an ordered trajectory, GDT effectively describes the evolution of regulatory relationships, thereby providing generalizable inter-genic features for universal GRN inference.

Note that VVP and GDT characterize the regulatory dependency from complementary perspectives, where VVP measures the response over value intervals, and GDT reflects instantaneous local sensitivities. To leverage their combined strengths, we implement a straightforward Ensemble (Ens) strategy by averaging their predicted logits, thereby producing a comprehensive and more generalizable description for regulatory relationships.

## 4. Experiments

### 4.1. Datasets

**GRN datasets.** We evaluate our method on seven scRNA-seq datasets from the BEELINE framework (Pratapa et al., 2020) spanning human (hESC, hHEP) and mouse (mDC, mESC, mHSC lineages) cell types, benchmarked against ground-truth GRNs derived from four distinct categories of biological evidence. Following the standard preprocessing pipeline, we focus on interactions outgoing from transcription factors (TFs). It is important to note that our method predicts relationships restricted to these TF-outgoing pairs as dictated by the evaluation protocol, rather than the model inherently learning directed edges from scratch. For each dataset, we select the top 500 and 1,000 highly variable genes (Bonferroni-corrected variance $P < 0.01$) (Stuart et al., 2019) alongside all identified TFs to construct the target networks. These two dataset configurations are hereafter referred to as TFs+500 and TFs+1000, respectively. All datasets are available via GEO.

**UGRN settings.** The core objective of UGRN inference is to assess the cross-dataset generalizability of the learned projector. Unlike conventional supervised methods that train and test on held-out pairs within the same dataset (sharing identical gene sets and expression matrix), we adopt a *Leave-One/Some-Dataset-Out* protocol. Formally, given the collection of GRN datasets $\mathbb{D} = \{\mathcal{D}_1, \ldots, \mathcal{D}_N\}$, we iterate through each dataset $\mathcal{D}_k \in \mathbb{D}$ (or a group of datasets for ablation studies) serving as the training source $\mathcal{D}_{tr}$. The trained projector $f_\phi$ is then evaluated on the remaining datasets. Note that the scFMs we applied follow scGPT and scbenchmark, sharing the same vocabulary with a size of 60697. This vocabulary covers 89.97% of the gene sets across the different datasets (on average). The proportion of genes shared between the training and test sets is 9.51% (on average), and the average label overlap between the train and

*Table 1.* AUPRC performance comparison on the UGRN benchmark employing scGPT and scBenchmark as frozen scFMs. The trainable parameters $f_\phi$ are optimized on the hESC dataset (from the STRING network), while evaluation is performed on all other heterogeneous datasets except hESC. Abbreviations: STR (STRING network), NSP (Non-Specific ChIP-seq network), SPC (Cell-type Specific ChIP-seq network), L/G (Literature/Knockdown network), and Origin (the original, unmodified baseline methods). Bold indicates the best performance, and underlined denotes the runner-up.

| | | scGPT | | | | | | | scBenchmark | | | | | | |
| | | ORIGIN | | BASELINE | | OURS | | | ORIGIN | | BASELINE | | OURS | | |
| NET | DATA | PERT | ATTN | PERT | EMB | VVP | GDT | ENS | PERT | ATTN | PERT | EMB | VVP | GDT | ENS |
|------|------|-------|-------|-------|-------|-------|-------|-------|-------|-------|-------|-------|-------|-------|-------|
| STR | HHEP | 0.496 | 0.507 | 0.586 | 0.732 | 0.609 | 0.906 | 0.909 | 0.537 | 0.606 | 0.591 | 0.738 | 0.668 | 0.912 | **0.927** |
| | MDC | 0.512 | 0.536 | 0.569 | 0.637 | 0.606 | 0.917 | 0.923 | 0.551 | 0.690 | 0.571 | 0.683 | 0.680 | 0.927 | **0.928** |
| | MESC | 0.542 | 0.531 | 0.493 | 0.699 | 0.600 | 0.969 | 0.966 | 0.567 | 0.617 | 0.496 | 0.693 | 0.666 | **0.974** | 0.966 |
| | MH-E | 0.550 | 0.496 | 0.592 | 0.757 | 0.674 | 0.843 | 0.868 | 0.560 | 0.645 | 0.599 | 0.766 | 0.722 | 0.835 | **0.879** |
| | MH-G | 0.573 | 0.500 | 0.597 | 0.805 | 0.706 | 0.826 | 0.845 | 0.608 | 0.701 | 0.631 | 0.797 | 0.782 | 0.819 | **0.907** |
| | MH-L | 0.622 | 0.534 | 0.624 | 0.815 | 0.656 | 0.895 | 0.873 | 0.655 | 0.764 | 0.697 | 0.784 | 0.809 | 0.852 | **0.934** |
| NSP | HHEP | 0.516 | 0.512 | 0.546 | 0.586 | 0.549 | 0.716 | 0.711 | 0.531 | 0.560 | 0.546 | 0.581 | 0.576 | **0.722** | 0.721 |
| | MDC | 0.546 | 0.529 | 0.553 | 0.606 | 0.579 | 0.787 | 0.795 | 0.538 | 0.644 | 0.552 | 0.610 | 0.624 | 0.801 | **0.804** |
| | MESC | 0.551 | 0.539 | 0.512 | 0.638 | 0.582 | 0.835 | **0.836** | 0.555 | 0.590 | 0.514 | 0.618 | 0.618 | 0.834 | 0.834 |
| | MH-E | 0.533 | 0.505 | 0.580 | 0.688 | 0.587 | 0.726 | 0.737 | 0.556 | 0.580 | 0.595 | 0.672 | 0.658 | 0.725 | **0.767** |
| | MH-G | 0.542 | 0.511 | 0.586 | 0.715 | 0.626 | 0.740 | 0.732 | 0.574 | 0.601 | 0.617 | 0.683 | 0.710 | 0.709 | **0.795** |
| | MH-L | 0.570 | 0.529 | 0.599 | 0.627 | 0.570 | 0.688 | 0.672 | 0.569 | 0.608 | 0.624 | 0.662 | 0.671 | 0.650 | **0.713** |
| L/G | MESC | 0.578 | 0.568 | 0.568 | **0.584** | 0.512 | 0.571 | 0.537 | 0.566 | 0.576 | 0.563 | 0.578 | 0.570 | 0.567 | 0.576 |
| SPC | HHEP | 0.518 | 0.560 | 0.525 | 0.556 | **0.603** | 0.529 | 0.557 | 0.559 | 0.549 | 0.528 | 0.567 | 0.580 | 0.509 | 0.571 |
| | MDC | 0.524 | 0.507 | 0.541 | 0.518 | 0.551 | 0.561 | 0.562 | 0.515 | **0.567** | 0.545 | 0.552 | 0.555 | 0.539 | 0.559 |
| | MESC | 0.644 | 0.647 | 0.632 | 0.635 | **0.678** | 0.626 | 0.661 | 0.651 | 0.650 | 0.636 | 0.649 | 0.673 | 0.643 | 0.673 |
| | MH-E | 0.675 | 0.668 | 0.682 | 0.676 | **0.701** | 0.639 | 0.639 | 0.681 | 0.683 | 0.683 | 0.681 | **0.701** | 0.659 | 0.695 |
| | MH-G | 0.689 | 0.672 | 0.679 | 0.686 | **0.726** | 0.643 | 0.656 | 0.677 | 0.696 | 0.678 | 0.679 | 0.684 | 0.682 | 0.686 |
| | MH-L | 0.635 | 0.653 | 0.654 | 0.621 | 0.656 | 0.638 | 0.638 | 0.652 | 0.653 | 0.663 | 0.603 | 0.658 | **0.674** | 0.660 |
| AVG | | 0.569 | 0.553 | 0.585 | 0.662 | 0.619 | 0.740 | 0.743 | 0.584 | 0.630 | 0.596 | 0.663 | 0.663 | 0.738 | **0.768** |

test splits is only 1.12%.

### 4.2. Implementation Details

For scFMs, we utilized the official implementations of scGPT (Cui et al., 2024) and scBenchmark (Qi et al., 2025). These serve as base pre-trained models with binned and continuous inputs, respectively. We implement the UGRN projector $f_\phi$ as an MLP to map the scFM-derived features to regulatory probabilities. The MLP consists of two hidden layers with dimensions 128 and 64, respectively, with ReLU activation functions, where the input size is defined by the specific feature extraction method. The final layer projects the latent representation to a scalar score bounded by a Sigmoid function. The model is trained using the Adam (Kingma & Ba, 2015) optimizer with a learning rate of $1 \times 10^{-3}$ and a batch size of 128 for 50 epochs, minimizing the binary cross-entropy loss. These hyperparameters are consistent across all methods to ensure fairness.

To capture comprehensive gene relationships, for all methods, we explicitly incorporate information from both directions when constructing the pair-wise feature $\mathbf{e}_{ij}$. Specifi-

cally, for a given pair, we concatenate the extracted vectors for both the forward and reverse directions. When implementing scGPT, by assigning virtual values to input gene expressions, we bypass the binning operation, which otherwise blocks gradient backpropagation, thereby enabling the calculation of gradients for genes. All hyperparameters are applied uniformly to all settings and datasets for UGRN, without specific tuning for individual settings. More details about the hyperparameters and additional ablation studies can be found in the Appendix.

### 4.3. Result Analysis

To provide deeper insights into the effectiveness of the UGRN setting and proposed methods, we organize our experiments to address the following core questions.

*Q1: Is the proposed Universal GRN setting rational?*

As shown in Table 1, the baseline Emb method, which maps pre-trained gene embeddings to regulatory links, consistently outperforms the original scFM methods (Origin-Pert and Origin-Attn) across most datasets. For example,

*Table 2.* Extended AUPRC comparison using scBenchmark across different UGRN settings. Each row represents the performance of a model trained on the specified dataset, reported as the average AUPRC calculated across all other datasets (excluding the training source and its network variants), e.g., the Str hESC row corresponds to the average performance reported in Table 1. "Need Exp" indicates whether input expression data is required for inference. The full AUROC performance could be found in the Appendix.

| Method | Origin | | Baseline | | Ours | | |
|---|---|---|---|---|---|---|---|
| Need Exp | Yes | Yes | Yes | No | No | No | No |
| Net Data | Pert | Attn | Pert | Emb | VVP | GDT | Ens |
| Str hESC | 0.584 | 0.630 | 0.596 | 0.663 | 0.663 | 0.738 | **0.768** |
| Str hHEP | 0.583 | 0.628 | 0.614 | 0.671 | 0.655 | 0.730 | **0.756** |
| Str mDC | 0.585 | 0.619 | 0.586 | 0.630 | 0.645 | 0.713 | **0.739** |
| Str mESC | 0.576 | 0.623 | 0.606 | 0.666 | 0.659 | 0.722 | **0.746** |
| Str mH-E | 0.574 | 0.618 | 0.588 | 0.656 | 0.644 | 0.723 | **0.743** |
| Str mH-G | 0.571 | 0.614 | 0.592 | 0.640 | 0.637 | 0.713 | **0.716** |
| Str mH-L | 0.570 | 0.612 | 0.571 | 0.627 | 0.615 | 0.703 | **0.712** |
| Nsp hESC | 0.584 | 0.630 | 0.569 | 0.616 | 0.632 | 0.721 | **0.733** |
| Nsp hHEP | 0.583 | 0.628 | 0.576 | 0.629 | 0.634 | 0.726 | **0.737** |
| Nsp mDC | 0.585 | 0.619 | 0.541 | 0.619 | 0.633 | 0.711 | **0.728** |
| Nsp mESC | 0.576 | 0.623 | 0.602 | 0.657 | 0.655 | 0.718 | **0.748** |
| Nsp mH-E | 0.574 | 0.618 | 0.578 | 0.640 | 0.642 | 0.714 | **0.732** |
| Nsp mH-G | 0.571 | 0.614 | 0.586 | 0.624 | 0.622 | 0.711 | **0.717** |
| Nsp mH-L | 0.570 | 0.612 | 0.575 | 0.618 | 0.597 | 0.678 | **0.685** |
| L/G mESC | 0.576 | 0.623 | 0.514 | 0.515 | 0.588 | 0.665 | **0.668** |
| Spc hESC | 0.584 | 0.630 | 0.566 | 0.549 | 0.604 | **0.640** | 0.639 |
| Spc hHEP | 0.583 | 0.628 | 0.573 | 0.569 | 0.604 | 0.672 | **0.674** |
| Spc mDC | 0.585 | 0.619 | 0.531 | 0.555 | 0.596 | **0.630** | 0.624 |
| Spc mESC | 0.576 | 0.623 | 0.546 | 0.563 | 0.615 | **0.677** | 0.665 |
| Spc mH-E | 0.574 | 0.618 | 0.565 | 0.599 | 0.578 | **0.655** | 0.621 |
| Spc mH-G | 0.571 | **0.614** | 0.557 | 0.596 | 0.571 | 0.597 | 0.576 |
| Spc mH-L | 0.570 | 0.612 | 0.557 | 0.602 | 0.577 | **0.676** | 0.645 |
| Avg | 0.577 | 0.621 | 0.572 | 0.614 | 0.621 | 0.692 | **0.699** |

*Table 3.* Sensitivity study on cross-dataset generalization performance using scBenchmark. $f_\phi$ is trained on the specific source dataset (indicated in each row) and evaluated on all other heterogeneous datasets. The reported results represent the average AUPRC on the test sets. Bold indicates the best performance.

| | Origin | | Baseline | | Ours | | |
|---|---|---|---|---|---|---|---|
| Data | Pert | Attn | Pert | Emb | VVP | GDT | Ens |
| hESC | 0.584 | 0.631 | 0.594 | 0.653 | 0.655 | 0.730 | **0.772** |
| hHEP | 0.583 | 0.628 | 0.617 | 0.659 | 0.656 | 0.736 | **0.771** |
| mDC | 0.585 | 0.619 | 0.585 | 0.645 | 0.641 | 0.719 | **0.736** |
| mESC | 0.576 | 0.623 | 0.588 | 0.649 | 0.646 | 0.732 | **0.763** |
| mH-E | 0.574 | 0.618 | 0.587 | 0.639 | 0.633 | 0.717 | **0.730** |
| mH-G | 0.571 | 0.614 | 0.583 | 0.636 | 0.607 | **0.717** | 0.715 |
| mH-L | 0.570 | 0.612 | 0.561 | 0.625 | 0.614 | 0.696 | **0.701** |
| Avg | 0.578 | 0.621 | 0.588 | 0.644 | 0.636 | 0.721 | **0.741** |

*Table 4.* Sensitivity study on cross-species generalization performance. $f_\phi$ is trained on the species indicated in each row and evaluated on the dataset of the other species. The reported results represent the average AUPRC on the test sets.

| | Origin | | Baseline | | Ours | | |
|---|---|---|---|---|---|---|---|
| Data | Pert | Attn | Pert | Emb | VVP | GDT | Ens |
| Human | 0.539 | 0.565 | 0.640 | 0.730 | 0.723 | 0.858 | **0.922** |
| Mouse | 0.592 | 0.642 | 0.570 | 0.704 | 0.666 | 0.923 | **0.936** |

on the scBenchmark, Baseline-Emb achieves an average AUPRC of 0.663, significantly surpassing the 0.630 average of Origin-Attn and the 0.584 average of Origin-Pert. This performance gap confirms our hypothesis that scFMs possess rich and generalizable latent biological priors. By adopting the UGRN setting, we can effectively bridge the gap between pre-trained representations of scFMs and explicit gene regulatory network inference, enabling better generalization to unseen genes and heterogeneous datasets than traditional implementations.

*Q2: Are the proposed methods effective?*

As shown in Table 1, our proposed methods demonstrate superior effectiveness across various datasets. On scBenchmark, our Ensemble (Ens) method achieves a state-of-the-art average AUPRC of 0.768, establishing a new benchmark for the task. This dominance is consistent across all categories; for instance, on the Str mH-L dataset, our method reaches a remarkable AUPRC of 0.934, significantly outperforming the strong Baseline-Emb's 0.784. Notably, the GDT method

alone reaches an average AUPRC of 0.738, outperforming the strong Baseline-Emb (0.663) by a substantial margin. The advantage is even more pronounced when compared to naive perturbation methods, where our Ens method outperforms Origin-Pert (0.584) by approximately 31.5% and surpasses Baseline-Pert (0.596) by 28.8%. This superiority is also observed in results based on scGPT. These findings validate that our VVP and GDT methods successfully extract significantly richer and more generalizable inter-gene features from scFMs, far exceeding the capabilities of traditional simplistic strategies like zero-out.

Regarding more "Leave-one-out" experiments, as shown in Table 2, our method maintains a dominant position across different UGRN settings where varying datasets are used for training. We achieve an average AUPRC of 0.699, significantly surpassing the averages of baselines. We observe that our method exhibits remarkable stability even in extreme cases where baselines collapse. For example, in the L/G mESC setting, Baseline-Emb fails to generalize, yielding an AUPRC of 0.515 (near random guessing), whereas our method recovers a high performance of 0.668, an improvement of over 34%. Even in the most challenging Spc network settings, our method consistently matches or exceeds the baselines. This indicates that our methods effectively extract intrinsic regulatory mechanisms that are invariant

across datasets.

*Q3: Are the proposed methods robust?*

To assess the robustness of our methods, we evaluate generalization capabilities across distinct data sources (Table 3), species (Table 4), and networks (Table 5). As shown in Table 3, our Ens demonstrates exceptional transferability in cross-dataset settings, achieving an average AUPRC of 0.741 compared to 0.644 for the strong Baseline-Emb. In cross-species settings, Ens achieves up to 0.936 AUPRC, substantially outperforming all baselines. Furthermore, Table 5 highlights the stability of our methods against network variations. In challenging settings such as the Spc network, where Baseline-Emb performance collapses to 0.540, our GDT method maintains 0.702. These results confirm that our methods effectively probe intrinsic and generalizable inter-gene mechanisms, ensuring robust prediction even under various settings with significant distribution shifts. To further ensure our model's performance is not artificially inflated by overlapping data, we conducted strict-gene and strict-label ablation studies. The results (detailed in Appendix) demonstrate that our methods maintain robust generalization even when overlapping genes and labels are strictly excluded from the training phase.

*Q4: How do our methods perform under imbalanced regulatory scenarios?*

While our primary experiments apply a balanced 1:1 ratio, we conduct additional experiments to investigate robustness by varying the Negative-to-Positive (N/P) ratio from 1 to 10 in Table 6. As the prevalence of positive samples becomes increasingly sparse, AUPRC naturally decreases. However, the AUROC scores remain remarkably stable, indicating that the intrinsic ranking capability is preserved. Crucially, our methods exhibit significantly higher resilience to sparsity compared to baselines. Under the extreme N/P ratio of 10, our Ens method outperforms Baseline-Emb by 56.4% (0.413 compared to 0.264). This relative margin is substantially larger than the 15.8% improvement observed in the balanced setting (0.768 compared to 0.663), demonstrating superior robustness in sparse scenarios.

### 4.4. Further Discussion

We established the UGRN setting to provide a rigorous benchmark for evaluating the extent to which features extracted by scFMs retain generalizable regulatory semantics. Our results validate UGRN settings and confirm that scFMs possess profound latent knowledge regarding gene regulation. We remain cognizant of the fact that real-world gene regulation is strictly context-dependent, heavily influenced by diverse biological covariates such as tissue ontology and the cellular microenvironment (Sonawane et al., 2017; Browaeys et al., 2020). While our current work isolates

*Table 5.* Sensitivity study on cross-network generalization performance. $f_\phi$ is trained on the specific source network and evaluated on datasets from the other networks. The reported results represent the average AUPRC on the test sets.

| NET | ORIGIN | | BASELINE | | OURS | | |
| --- | --- | --- | --- | --- | --- | --- | --- |
| | PERT | ATTN | PERT | EMB | VVP | GDT | ENS |
| STR | 0.579 | 0.602 | 0.605 | 0.667 | 0.627 | 0.644 | **0.676** |
| NSP | 0.591 | 0.635 | 0.625 | 0.679 | 0.662 | 0.719 | **0.743** |
| L/G | 0.578 | 0.623 | 0.518 | 0.521 | 0.588 | **0.678** | 0.675 |
| SPC | 0.564 | 0.622 | 0.535 | 0.540 | 0.606 | **0.702** | 0.672 |
| AVG | 0.578 | 0.620 | 0.571 | 0.602 | 0.621 | 0.686 | **0.692** |

*Table 6.* Sensitivity study on the impact of class imbalance. We report both AUROC and AUPRC performance under varying Negative-to-Positive (N/P) ratios in the training/test set. The AUPRC N/P=1 row corresponds to the average performance reported in Table 1. Bold indicates the best performance.

| | N/P | BASELINE | | OURS | | |
| --- | --- | --- | --- | --- | --- | --- |
| | | PERT | EMB | VVP | GDT | ENS |
| AUROC | 1 | 0.543 | 0.608 | 0.612 | 0.664 | **0.704** |
| | 2 | 0.547 | 0.600 | 0.607 | 0.654 | **0.705** |
| | 3 | 0.556 | 0.595 | 0.604 | 0.662 | **0.705** |
| | 5 | 0.556 | 0.590 | 0.609 | 0.667 | **0.705** |
| | 10 | 0.558 | 0.589 | 0.604 | 0.678 | **0.706** |
| | AVG | 0.552 | 0.596 | 0.607 | 0.665 | **0.705** |
| AUPRC | 1 | 0.596 | 0.663 | 0.663 | 0.738 | **0.768** |
| | 2 | 0.450 | 0.522 | 0.518 | 0.630 | **0.673** |
| | 3 | 0.377 | 0.437 | 0.434 | 0.572 | **0.614** |
| | 5 | 0.292 | 0.352 | 0.346 | 0.497 | **0.536** |
| | 10 | 0.208 | 0.264 | 0.247 | 0.394 | **0.413** |
| | AVG | 0.385 | 0.448 | 0.442 | 0.566 | **0.601** |

the underlying invariant mechanism of regulation, future applications should integrate these context variables into the UGRN framework to model complex, environment-specific GRN inference.

Unlike traditional scFM applications that rely on direct outputs (e.g., attention weights), our approach utilizes a labeled dataset to train a "translator". While this may appear to relax the zero-shot constraint on the genes from the training set, it remains strictly zero-shot regarding the inference on unseen genes. By leveraging a limited set of known interactions as a biological prior, we successfully align the scFM's latent representation space with the gene regulation. This implies that realizing UGRN inference relies on a combination of pre-trained scFMs and generalizable priors, a challenge that warrants significant attention in future investigations.

## 5. Conclusion

In this paper, we identified and addressed the critical misalignment between the reconstruction-based pre-training of Single-cell Foundation Models (scFMs) and the implicit inter-gene relationship requirements of GRN inference. To bridge this gap, we established the Universal Gene Regulatory Network (UGRN) framework, shifting the paradigm from dataset-specific fitting to the learning of generalizable regulatory features, thereby demonstrating that these pretrained models encode generalizable inter-gene representations. Furthermore, we proposed Virtual Value Perturbation (VVP) and Gradient Trajectory (GDT). These methods decouple feature extraction from observational expressions, allowing us to distill intrinsic regulatory logic from frozen scFMs through virtual queries, successfully extracting deep, generalizable features inherent in scFMs. Extensive experiments validate the rationality of the UGRN setting, demonstrating that our methods significantly outperform both original scFM applications and strong baselines. Notably, our approach exhibits exceptional resilience, where it maintains high performance in challenging cross-species and cross-network scenarios where baseline methods collapse, and demonstrates more stability under class imbalance. These findings confirm that scFMs encode profound, generalizable biological priors and that our methods effectively unlock this potential to establish a solid foundation for Universal GRN Inference.

## Acknowledgments

This work was supported by the Strategic Priority Research Program of the Chinese Academy of Sciences under Grant No. XDA0460205.

## Impact Statement

This paper presents work aimed at advancing the field of single-cell foundation models. By enabling Universal Gene Regulatory Network inference, our framework has the potential to accelerate the identification of disease mechanisms and therapeutic targets, particularly in scenarios lacking extensive observational data. While this work facilitates biological discovery, regulatory links predicted by our models are computational hypotheses that require experimental validation before clinical application. We do not foresee immediate negative ethical or societal consequences specifically arising from this work.

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

# A. Detailed Methodological Formulations

In this section, we present the detailed algorithmic formulation of the UGRN framework, encompassing both the Leave-One-Dataset-Out training protocol and the universal inference mechanism. We specifically outline the construction of our proposed feature extractors, Virtual Value Perturbation (VVP) and Gradient Trajectory (GDT).

---

**Algorithm 1** UGRN inference: Training and Universal Inference

---

1: **Input:** Datasets $\mathbb{D}$, scFM $\mathcal{M}$, Hyperparams: $v_b$, $\{v_{p,1}, \ldots, v_{p,M}\}$, $\{v_{b,1}, \ldots, v_{b,T}\}$.
2: **Output:** Trained Projectors $f_\phi$, UGRN Score $s_{uv}$.
3: **Stage 1: Training (Leave-One-Dataset-Out)**
4: Partition $\mathbb{D}$ into $\mathcal{D}_{tr}$ and held-out $\mathcal{D}_{test}$. Initialize $\phi_{vvp}, \phi_{gdt}$.
5: **while** not converged **do**
6:     Sample gene pair batch $(g_i, g_j)$ from $\mathcal{D}_{tr}$.
7:     *Feature Extraction using frozen $\mathcal{M}$:*
8:         $\mathbf{e}_{vvp} \leftarrow$ Compute VVP features using base $v_b$ and targets $\{v_{p,m}\}$ via Eq. (8).
9:         $\mathbf{e}_{gdt} \leftarrow$ Compute GDT features using trajectory set $\{v_{b,t}\}$ via Eq. (9).
10:     Update $\phi_{vvp}, \phi_{gdt}$ by minimizing supervision loss, i.e., Eq. (2).
11: **end while**
12: ────────────────────────────────────
13: **Stage 2: Universal Inference**
14: **Given:** Arbitrary source $g_u$ and target $g_v$ from $\mathcal{D}_{test}$.
15: Perform feature extraction for $(g_u, g_v)$ using frozen $\mathcal{M}$:
16:     Get $\mathbf{e}_{vvp}$ using base $v_b$ and targets $\{v_{p,m}\}$.
17:     Get $\mathbf{e}_{gdt}$ using trajectory set $\{v_{b,t}\}$.
18: **return** UGRN Score $s_{uv} = \frac{1}{2}(f_{\phi_{vvp}}(\mathbf{e}_{vvp}) + f_{\phi_{gdt}}(\mathbf{e}_{gdt}))$.

---

# B. Dataset Descriptions

## B.1. Data Sources and Statistics

### Datasets and Data Sources

We evaluate our method on seven benchmark scRNA-seq datasets curated by the BEELINE framework (Pratapa et al., 2020), covering a diverse range of human and mouse biological systems. The datasets include human embryonic stem cells (hESC), human hepatocytes (hHEP), mouse embryonic stem cells (mESC), mouse dendritic cells (mDC), and three lineages of mouse hematopoietic stem cells (mHSC-E, mHSC-GM, and mHSC-L). All datasets are publicly available from GEO: GSE75748 (hESC), GSE81252 (hHEP), GSE98664 (mESC), GSE48968 (mDC), and GSE81682 (mHSC variants). Each dataset is processed to include highly variable transcription factors, evaluated under two distinct scales: TFs+500 and TFs+1000, corresponding to the top 500 and 1000 TFs respectively ranked by expression variability.

### Ground-Truth Networks and Experimental Setup

To provide supervision for GRN inference, we utilize four types of ground-truth regulatory networks derived from multiple sources of biological evidence: (1) *STRING networks* (Szklarczyk et al., 2019), constructed from protein-protein interaction databases; (2) *non-cell-type-specific ChIP-seq networks (Non-Specific)* (Liu et al., 2015; Garcia-Alonso et al., 2019), built from TF binding profiles aggregated across diverse contexts; (3) *cell-type-specific ChIP-seq (Specific)* networks (Xu et al., 2013; Moore et al., 2020), which offer high-resolution, context-specific TF–target interactions; and (4) *LOF/GOF networks* (Xu et al., 2013), based on experimentally validated perturbation-derived causal interactions (specifically for mESC). Regarding network characteristics, the Specific networks typically contain more positive TF–target pairs, resulting in more balanced datasets. In contrast, STRING and Non-Specific networks are significantly sparser, often presenting highly imbalanced classification scenarios. To ensure statistical robustness and reliability, all reported results are averaged over ten independent trials initialized with different random seeds. For a comprehensive breakdown of the dataset characteristics and ground-truth network statistics under both TFs-500 and TFs-1000 settings, please refer to Table A1 and Table A2.

*Table A1.* Statistics of the seven benchmark datasets under the TFs+500 setting. This table summarizes the number of genes and cells, along with network properties (transcription factors, targets, positive edges, and density) derived from four distinct ground-truth sources: STRING, Non-Specific, Specific, and LOF/GOF.

| Dataset | Genes | Cells | STRING | | | | Non-Specific | | | | Specific | | | | LOF/GOF | | | |
|---|---|---|---|---|---|---|---|---|---|---|---|---|---|---|---|---|---|---|
| | | | TFs | Targets | Pos. | Dens. | TFs | Targets | Pos. | Dens. | TFs | Targets | Pos. | Dens. | TFs | Targets | Pos. | Dens. |
| hESC | 910 | 758 | 343 | 511 | 4257 | 0.024 | 283 | 753 | 3441 | 0.016 | 34 | 815 | 4545 | 0.164 | - | - | - | - |
| hHEP | 948 | 425 | 409 | 646 | 7523 | 0.028 | 322 | 825 | 4129 | 0.015 | 30 | 874 | 9939 | 0.379 | - | - | - | - |
| mDC | 821 | 383 | 264 | 479 | 4815 | 0.038 | 250 | 634 | 3067 | 0.019 | 20 | 443 | 756 | 0.085 | - | - | - | - |
| mESC | 1120 | 421 | 495 | 638 | 7762 | 0.024 | 516 | 890 | 6893 | 0.015 | 88 | 977 | 29613 | 0.345 | 34 | 774 | 4169 | 0.158 |
| mHSC-E | 704 | 1071 | 156 | 291 | 1371 | 0.029 | 144 | 442 | 1425 | 0.022 | 23 | 691 | 11557 | 0.578 | - | - | - | - |
| mHSC-GM | 632 | 889 | 92 | 201 | 748 | 0.040 | 82 | 297 | 743 | 0.030 | 22 | 618 | 7364 | 0.543 | - | - | - | - |
| mHSC-L | 560 | 847 | 39 | 70 | 137 | 0.048 | 35 | 164 | 279 | 0.048 | 16 | 525 | 4398 | 0.525 | - | - | - | - |

*Table A2.* Statistics of the seven benchmark datasets under the TFs+1000 setting. This table summarizes the number of genes and cells, along with network properties (transcription factors, targets, positive edges, and density) derived from four distinct ground-truth sources: STRING, Non-Specific, Specific, and LOF/GOF.

| Dataset | Genes | Cells | STRING | | | | Non-Specific | | | | Specific | | | | LOF/GOF | | | |
|---|---|---|---|---|---|---|---|---|---|---|---|---|---|---|---|---|---|---|
| | | | TFs | Targets | Pos. | Dens. | TFs | Targets | Pos. | Dens. | TFs | Targets | Pos. | Dens. | TFs | Targets | Pos. | Dens. |
| hESC | 1410 | 758 | 351 | 695 | 5149 | 0.021 | 292 | 1138 | 4617 | 0.014 | 34 | 1260 | 7084 | 0.165 | - | - | - | - |
| hHEP | 1448 | 425 | 414 | 874 | 9003 | 0.024 | 332 | 1331 | 5351 | 0.013 | 31 | 1331 | 15558 | 0.377 | - | - | - | - |
| mDC | 1321 | 383 | 273 | 664 | 5898 | 0.032 | 254 | 969 | 3918 | 0.016 | 21 | 684 | 1193 | 0.082 | - | - | - | - |
| mESC | 1620 | 421 | 499 | 785 | 8479 | 0.021 | 522 | 1214 | 8030 | 0.013 | 89 | 1385 | 42795 | 0.347 | 34 | 1098 | 5742 | 0.154 |
| mHSC-E | 1204 | 1071 | 161 | 413 | 1826 | 0.027 | 147 | 674 | 1960 | 0.020 | 29 | 1177 | 21975 | 0.566 | - | - | - | - |
| mHSC-GM | 1132 | 889 | 100 | 344 | 1311 | 0.037 | 88 | 526 | 1358 | 0.029 | 23 | 1089 | 14135 | 0.561 | - | - | - | - |
| mHSC-L | 692 | 847 | 40 | 81 | 154 | 0.045 | 37 | 192 | 317 | 0.043 | 16 | 640 | 5180 | 0.507 | - | - | - | - |

## B.2. More Implementation Details

For our methods, we adopt the following hyperparameters based on empirical experience. For continuous inputs, for Virtual Value Perturbation, we set the virtual base value $v_b = 2.0$ and employ 10 virtual target values $v_p$, uniformly distributed across the expression range $[0, 30]$. For Gradient Trajectory, we compute the gradient trajectory for different $v_b$ over the interval $[0, 4.0]$ with a step size of 0.1 to capture fine-grained sensitivity. For scGPT (binned inputs), we set the virtual base value $v_b = 26$ and employ 13 virtual target values $v_p$, uniformly distributed across $[0, 130]$ in VVP and set $v_b$ over the interval $[0, 32]$ with 8 geometric steps in GDT. Note that thanks to our designed virtual values, we do not need to conduct the binning operation, which allows the gradient backpropagation. These hyperparameters are applied uniformly to all settings and datasets for UGRN, without specific tuning for individual settings.

## B.3. Train/Test Datasets for Universal GRN Setting

To rigorously assess the cross-gene generalizability of our framework, we employ a *Leave-One/Group-Out* protocol that strictly separates training and testing sets, differing fundamentally from conventional intra-dataset cross-validation. Formally, given a collection of datasets $\mathbb{D}$, we partition them into disjoint source $\mathcal{D}_{tr}$ and target $\mathcal{D}_{test}$ sets to simulate realistic discovery scenarios. In our main experiments, we iterate through each dataset as the held-out target while training on the remaining sources. For ablation studies, we extend this to coarser granularities, such as partitioning by species (e.g., training on *Mouse* data to predict on *Human*) or network annotation sources to verify robustness against specific experimental settings. This setup ensures a strict zero-shot evaluation where the model is prohibited from accessing ground-truth labels from the target domain, forcing it to learn transferable regulatory semantics rather than memorizing dataset-specific correlations. Importantly, by leveraging the pre-trained vocabulary of the underlying scFM, this protocol enables *Universal GRN Inference*, unlike traditional methods confined to the intersection of training gene sets, our trained projector can infer regulatory scores for any arbitrary gene pair $(g_u, g_v)$ present in the scFM's token dictionary, regardless of their absence in $\mathcal{D}_{tr}$.

## C. Further Ablation Experiments

**Full AUROC Performance on TFs+500.** We report AUROC performance on the TFs+500 setting, to complement the Table 2 in the main paper. As shown in Table A3, the results show that our method consistently achieves superior performance

*Table A3.* Full AUROC comparison using scBenchmark across different UGRN settings. All settings are the same as Table 2.

| METHOD | | ORIGIN | | BASELINE | | OURS | | |
|---|---|---|---|---|---|---|---|---|
| NEED EXP | | YES | YES | YES | NO | NO | NO | NO |
| NET | DATA | PERT | ATTN | PERT | EMB | VVP | GDT | ENS |
| | HESC | 0.530 | 0.585 | 0.543 | 0.608 | 0.612 | 0.664 | 0.704 |
| | HHEP | 0.526 | 0.581 | 0.574 | 0.616 | 0.603 | 0.652 | 0.690 |
| | MDC | 0.528 | 0.573 | 0.535 | 0.572 | 0.591 | 0.623 | 0.675 |
| STR | MESC | 0.528 | 0.583 | 0.561 | 0.625 | 0.611 | 0.645 | 0.684 |
| | MH-E | 0.525 | 0.580 | 0.556 | 0.618 | 0.597 | 0.655 | 0.682 |
| | MH-G | 0.525 | 0.574 | 0.552 | 0.595 | 0.594 | 0.641 | 0.656 |
| | MH-L | 0.520 | 0.573 | 0.524 | 0.581 | 0.577 | 0.631 | 0.646 |
| | HESC | 0.530 | 0.585 | 0.521 | 0.566 | 0.589 | 0.642 | 0.666 |
| | HHEP | 0.526 | 0.581 | 0.524 | 0.582 | 0.587 | 0.649 | 0.666 |
| | MDC | 0.528 | 0.573 | 0.487 | 0.571 | 0.583 | 0.627 | 0.657 |
| NSP | MESC | 0.528 | 0.583 | 0.560 | 0.617 | 0.608 | 0.639 | 0.682 |
| | MH-E | 0.525 | 0.580 | 0.543 | 0.605 | 0.595 | 0.641 | 0.667 |
| | MH-G | 0.525 | 0.574 | 0.552 | 0.577 | 0.585 | 0.638 | 0.654 |
| | MH-L | 0.520 | 0.573 | 0.532 | 0.573 | 0.553 | 0.613 | 0.620 |
| L/G | MESC | 0.528 | 0.583 | 0.456 | 0.439 | 0.542 | 0.604 | 0.608 |
| | HESC | 0.530 | 0.585 | 0.521 | 0.496 | 0.560 | 0.562 | 0.578 |
| | HHEP | 0.526 | 0.581 | 0.517 | 0.529 | 0.567 | 0.591 | 0.605 |
| | MDC | 0.528 | 0.573 | 0.474 | 0.500 | 0.547 | 0.572 | 0.571 |
| SPC | MESC | 0.528 | 0.583 | 0.498 | 0.533 | 0.568 | 0.598 | 0.606 |
| | MH-E | 0.525 | 0.580 | 0.533 | 0.580 | 0.536 | 0.577 | 0.568 |
| | MH-G | 0.525 | 0.574 | 0.515 | 0.579 | 0.532 | 0.538 | 0.540 |
| | MH-L | 0.520 | 0.573 | 0.517 | 0.577 | 0.532 | 0.604 | 0.600 |
| AVG | | 0.526 | 0.579 | 0.527 | 0.570 | 0.576 | 0.618 | 0.638 |

*Table A4.* Aggregated performance comparison on scBenchmark on TFs+1000 settings (Further experiments for Table 2 in the main paper). "Need Exp" indicates whether real expression data is required during inference.

| METHOD | | ORIGIN | | BASELINE | | OURS | | |
|---|---|---|---|---|---|---|---|---|
| NEED EXP | | YES | YES | YES | NO | NO | NO | NO |
| NET | DATA | PERT | ATTN | PERT | EMB | VVP | GDT | ENS |
| | HESC | 0.585 | 0.632 | 0.594 | 0.664 | 0.659 | 0.706 | 0.733 |
| | HHEP | 0.584 | 0.629 | 0.611 | 0.675 | 0.651 | 0.709 | 0.736 |
| | MDC | 0.585 | 0.619 | 0.598 | 0.650 | 0.644 | 0.698 | 0.716 |
| STR | MESC | 0.576 | 0.622 | 0.608 | 0.671 | 0.653 | 0.700 | 0.720 |
| | MH-E | 0.575 | 0.619 | 0.587 | 0.653 | 0.642 | 0.702 | 0.722 |
| | MH-G | 0.575 | 0.615 | 0.593 | 0.644 | 0.639 | 0.700 | 0.708 |
| | MH-L | 0.572 | 0.613 | 0.583 | 0.626 | 0.621 | 0.681 | 0.693 |
| | HESC | 0.585 | 0.632 | 0.550 | 0.618 | 0.633 | 0.701 | 0.709 |
| | HHEP | 0.584 | 0.629 | 0.588 | 0.617 | 0.628 | 0.705 | 0.714 |
| | MDC | 0.585 | 0.619 | 0.599 | 0.623 | 0.632 | 0.688 | 0.701 |
| NSP | MESC | 0.576 | 0.622 | 0.601 | 0.651 | 0.648 | 0.699 | 0.725 |
| | MH-E | 0.575 | 0.619 | 0.567 | 0.647 | 0.632 | 0.696 | 0.713 |
| | MH-G | 0.575 | 0.615 | 0.588 | 0.637 | 0.641 | 0.697 | 0.708 |
| | MH-L | 0.572 | 0.613 | 0.568 | 0.610 | 0.615 | 0.656 | 0.666 |
| L/G | MESC | 0.576 | 0.622 | 0.522 | 0.523 | 0.623 | 0.648 | 0.673 |
| | HESC | 0.585 | 0.632 | 0.572 | 0.551 | 0.609 | 0.659 | 0.660 |
| | HHEP | 0.584 | 0.629 | 0.574 | 0.568 | 0.595 | 0.632 | 0.631 |
| | MDC | 0.585 | 0.619 | 0.562 | 0.551 | 0.606 | 0.623 | 0.631 |
| SPC | MESC | 0.576 | 0.622 | 0.543 | 0.569 | 0.626 | 0.626 | 0.629 |
| | MH-E | 0.575 | 0.619 | 0.569 | 0.591 | 0.558 | 0.553 | 0.567 |
| | MH-G | 0.575 | 0.615 | 0.570 | 0.591 | 0.564 | 0.599 | 0.573 |
| | MH-L | 0.572 | 0.613 | 0.558 | 0.596 | 0.588 | 0.637 | 0.617 |
| AVG | | 0.579 | 0.621 | 0.578 | 0.615 | 0.623 | 0.669 | 0.679 |

compared to both Origin and Baseline methods. The alignment between AUROC and AUPRC rankings further validates the robustness of our methods in distinguishing true regulatory interactions across various datasets and network structures.

*Table A5.* Ablation study on the base value $v_b$ for VVP. We compare different virtual base values. Test settings are the same as Table 1 in the main paper.

| BASE VALUE ($v_b$) | 0.0 | 0.5 | 1.0 | 1.5 | 2.0 | 2.5 | 3.0 | 3.5 | 4.0 | 4.5 |
|---|---|---|---|---|---|---|---|---|---|---|
| AVG. AUPRC | 0.612 | 0.611 | 0.618 | 0.643 | 0.663 | 0.662 | 0.656 | 0.648 | 0.643 | 0.636 |
| AVG. AUROC | 0.571 | 0.571 | 0.576 | 0.596 | 0.612 | 0.610 | 0.605 | 0.597 | 0.592 | 0.585 |

*Table A6.* Ablation study on the number of virtual target values $M$ for VVP. The default setting is $M = 10$. Test settings are the same as Table 1 in the main paper.

| NUM. TARGETS ($M$) | 2 | 4 | 6 | 8 | 10 | 12 | 14 | 16 | 18 | 20 |
|---|---|---|---|---|---|---|---|---|---|---|
| AVG. AUPRC | 0.590 | 0.655 | 0.660 | 0.661 | 0.663 | 0.658 | 0.662 | 0.660 | 0.663 | 0.661 |
| AVG. AUROC | 0.540 | 0.603 | 0.609 | 0.610 | 0.612 | 0.604 | 0.611 | 0.606 | 0.612 | 0.609 |

*Table A7.* Ablation study on the perturbation range (Max Value) for VVP. The default setting is Max=30. Test settings are the same as Table 1 in the main paper.

| RANGE MAX | 5 | 10 | 15 | 20 | 25 | 30 | 35 | 40 | 45 | 50 |
|---|---|---|---|---|---|---|---|---|---|---|
| AVG. AUPRC | 0.652 | 0.655 | 0.659 | 0.660 | 0.662 | 0.663 | 0.660 | 0.656 | 0.658 | 0.655 |
| AVG. AUROC | 0.604 | 0.607 | 0.609 | 0.610 | 0.611 | 0.612 | 0.610 | 0.605 | 0.609 | 0.604 |

**Performance on TFs+1000.** We extend our evaluation to the TFs+1000 setting, where the model is tested on a larger set of unseen transcription factors. As shown in Table A4, our method consistently outperforms baseline approaches across all settings. Importantly, the relative ranking remains consistent with the TFs+500 setting reported in the main paper. This confirms that our methods effectively generalize to novel regulatory mechanisms, maintaining SOTA performance.

**Hyperparameter Sensitivity Analysis.** To validate that our results are not the product of extensive hyperparameter tuning (e.g., exhaustive grid search), based on pre-trained scFM, scBenchmark, we conduct a sensitivity analysis on the components of Virtual Value Perturbation (VVP) and Gradient Trajectory (GDT). The results below demonstrate that our method is highly robust, where performance remains stable and near-optimal across a wide range of hyperparameter choices.

**Ablation on Virtual Value Perturbation (VVP).** We first analyze the initialization of the perturbation source value ($v_b$). Table A5 compares a zero vector and different virtual base values. While the base ($v_b = 2.0$) yields the best results to activate the scFM, the performance variation within the range $[2.0, 3.0]$ is minimal. This indicates that precise tuning of $v_b$ is unnecessary as long as the value avoids extreme out-of-distribution shifts (like $v_b = 0$).

Next, we examine the perturbation resolution ($M$) and range. Table A6 shows that performance saturates quickly. Once $M \geq 6$, the AUPRC fluctuates negligibly around 0.66. This suggests that a coarse resolution is sufficient to capture regulatory responses, and the model is not sensitive to the specific number of targets. Similarly, Table A7 shows a broad "safe" range for the perturbation magnitude (approx. 20–35). As long as the range is wide enough to capture gene saturation effects, the specific cutoff value has little impact on the final result.

**Ablation on Gradient Trajectory (GDT).** We assess the definition of the gradient trajectory path. Table A9 confirms that extending the aggregation interval captures broader dynamic sensitivity, with performance plateauing after an interval of 3.5. Any interval defined beyond this point yields consistently high performance.

Finally, Table A10 provides the strongest evidence for the robustness of our approach. We observe almost no variation in AUPRC (approx. 0.737) as the number of integration steps ($T$) varies from 20 to 60. This stability confirms that our results are not dependent on a specific, carefully tuned hyperparameter; rather, any reasonable sampling density is sufficient to approximate the regulatory dynamics effectively.

**Ablation on Data Overlap.** To ensure our performance is not artificially inflated by data leakage, we evaluate our methods under a strictly non-overlapping setting. While our standard evaluation utilizing the BEELINE framework inherently maintains a minimal overlap between training and testing splits (averaging only 1.12% for labels and 9.51% for genes), Table A8 confirms that explicitly eliminating all seen labels (Strict-label) has a negligible impact on overall performance. For instance, our Ensemble method (Ens) exhibits almost no variation, maintaining an AUPRC of 0.767 compared to 0.768 in the standard setting.

*Table A8.* AUPRC performance comparison under strict non-overlapping settings on scBenchmark. "Standard" corresponds to the average performance reported in Table 1. "Strict-label" and "Strict-gene" indicate evaluations where all overlapping labels or genes between the training and test sets are explicitly removed.

| SETTING | BASELINE | | OURS | | |
|---|---|---|---|---|---|
| | PERT | EMB | VVP | GDT | ENS |
| STANDARD | 0.596 | 0.663 | 0.663 | 0.738 | 0.768 |
| STRICT-LABEL | 0.591 | 0.645 | 0.658 | 0.735 | 0.767 |
| STRICT-GENE | 0.608 | 0.590 | 0.651 | 0.731 | 0.756 |

*Table A9.* Ablation study on the trajectory range for GDT. The default integration interval is $[0, 4.0]$. Test settings are the same as Table 1 in the main paper.

| INTERVAL MAX | 2.0 | 2.5 | 3.0 | 3.5 | 4.0 | 4.5 | 5.0 | 5.5 | 6.0 | 6.5 |
|---|---|---|---|---|---|---|---|---|---|---|
| AVG. AUPRC | 0.721 | 0.727 | 0.733 | 0.738 | 0.738 | 0.737 | 0.737 | 0.738 | 0.737 | 0.737 |
| AVG. AUROC | 0.639 | 0.648 | 0.657 | 0.663 | 0.665 | 0.663 | 0.662 | 0.665 | 0.664 | 0.665 |

*Table A10.* Ablation study on the sampling resolution (Steps $T$) for GDT. The default setting is $T = 40$ (Step size 0.1 over $[0, 4.0]$). Test settings are the same as Table 1 in the main paper.

| STEPS ($T$) | 15 | 20 | 25 | 30 | 35 | 40 | 45 | 50 | 55 | 60 |
|---|---|---|---|---|---|---|---|---|---|---|
| AVG. AUPRC | 0.731 | 0.737 | 0.736 | 0.736 | 0.737 | 0.738 | 0.737 | 0.738 | 0.738 | 0.738 |
| AVG. AUROC | 0.655 | 0.662 | 0.660 | 0.661 | 0.661 | 0.665 | 0.664 | 0.664 | 0.665 | 0.665 |

# D. Further Discussion and Limitations

## D.1. Broader Impact

*Restoring Confidence in scFMs for GRN inference.* Recent skepticism has arisen regarding the empirical utility of Single-Cell Foundation Models (scFMs) for downstream inference tasks, with some studies suggesting they barely outperform random baselines. Our work counters this narrative by demonstrating that scFMs do possess deep, transferable regulatory knowledge, but accessing it requires the right interrogation mechanisms (i.e., VVP and GDT) rather than simple proxies. By validating the efficacy of scFMs in the UGRN setting, we provide a renewed impetus for the community to leverage these powerful pre-trained models for biological discovery.

*Resource Efficiency and Accelerating Discovery.* Our proposed UGRN paradigm allows for the prediction of regulatory relationships between unseen genes without requiring new, costly sequencing experiments or model retraining. This has profound implications for drug target discovery and the understanding of disease mechanisms, as it allows researchers to generate high-quality hypotheses for unmeasured genes virtually. Furthermore, by utilizing frozen scFMs as universal feature extractors rather than training large-scale models from scratch for every new heterogeneous dataset, we significantly reduce the computational carbon footprint, aligning with the goals of Green AI.

*Expanding the Boundaries of GRN Research.* Finally, we formalize the Universal Gene Regulatory Network inference setting, shifting the focus from closed-world, dataset-specific inference to open-world generalization. This establishes a new rigorous benchmark and problem framework for the community, encouraging the development of methods that can disentangle fundamental regulatory logic from dataset-specific noise.

## D.2. Limitations

*Dependency on scFM Vocabulary.* The generalizability of our method is strictly upper-bounded by the gene vocabulary of the underlying scFM. If a specific gene was never encountered during the scFM's pre-training stage, the model cannot generate a valid embedding or extract meaningful features for the gene, rendering our method inapplicable to that specific gene. While this can be partly solved by Universal Cell Embeddings (Rosen et al., 2024), or theoretically mitigated by approximation methods, such as simulating an unseen gene's features via a weighted sum of homologous genes based on RNA sequence similarity, which does not guarantee that the model possesses true knowledge of the interaction of the gene with others. Consequently, the ultimate solution to this limitation relies on the continued evolution of scFMs to encompass broader and more diverse transcriptomic atlases.

## D.3. Further Discussion

*Impact of Positive-Negative Sampling Ratio.* In constructing our training and evaluation benchmarks, we adopt a balanced 1:1 positive-to-negative ratio. Standard GRN datasets typically provide only experimentally validated positive interactions, leaving non-interacting pairs implicit. Due to the incompleteness of current biological knowledge, treating all unobserved pairs as negatives would introduce a substantial number of false negatives. This label noise is particularly harmful in our UGRN setting, where the objective is to learn a generalizable $f_\phi$ rather than overfitting to dataset-specific statistics. By maintaining a 1:1 ratio via random sampling, we minimize the inclusion of false negatives, ensuring high-quality ground truth. Furthermore, as shown in our ablations, our method maintains its superiority across various class imbalance ratios, confirming that our performance gains are robust and not artifacts of this specific sampling strategy.

*Biological Context Specificity.* As discussed in the main paper, our UGRN setting assumes that genes share a foundational, universal regulatory logic that can be captured by a translator $f_\phi$. While this holds for fundamental interactions, biological reality is far more complex, where the same gene pair may exhibit different regulatory behaviors depending on the species, organ, or cellular micro-environment. Our current framework abstracts away these context-specific variations to achieve broad generalization. To be fully applicable to precision biological experiments, the UGRN framework would need to be augmented with context-aware parameters or condition-specific prompts, an area we leave for future exploration.

