# OpenReview forum: "Towards Universal Gene Regulatory Network Inference: Unlocking Generalizable Regulatory Knowledge in Single-cell Foundation Models"
_ICML.cc/2026/Conference — ICML 2026 regular_

### Official Review · Reviewer_Xch5 · 2026-03-06

**Soundness:** 4
**Presentation:** 4
**Significance:** 3
**Originality:** 3
**Overall Recommendation:** 5
**Confidence:** 5

**Summary:**

The paper provides a framework to improve existing scFM to perform GRN inference by introducing an additional nonlinear function to project the previous "in silico perturbation" results to the GRN prediction results. The paper is well writtern and the idea is interesting, although straightforward.  The authors also made many experiments to prove the results for many "unseen" cases.  I only have one major concern:

The results of the proposed GDT method provide a huge improvement over the baseline method in STR network (as much as 0.2-0.3 in AUPRC). I feel a little confused about it as the baseline method also uses the GRN dataset information. Is there any reason for it?

There are some minor suggestions for writing:
1. The meaning of several words in Section 4 is not mentioned. E.g., what is the Origin method in Table 1(I guess it means using scFM without a projection function). What is NSP, L/G, SPC NET?

2. The word "Ablation study" is not suitable for Table 3-5. Maybe use "Sensitivity study".

**Compliance With Llm Reviewing Policy:**

Affirmed.

**Final Justification:**

I decide to keep the score.

**Key Questions For Authors:**

I only have one major concern:

The results of the proposed GDT method provide a huge improvement over the baseline method in STR network (as much as 0.2-0.3 in AUPRC). I feel a little confused about it as the baseline method also uses the GRN dataset information. Is there any reason for it?

There are some minor suggestions for writing:
1. The meaning of several words in Section 4 is not mentioned. E.g., what is the Origin method in Table 1(I guess it means using scFM without a projection function). What is NSP, L/G, SPC NET?

2. The word "Ablation study" is not suitable for Table 3-5. Maybe use "Sensitivity study".

**Limitations:**

I only have one major concern:

The results of the proposed GDT method provide a huge improvement over the baseline method in STR network (as much as 0.2-0.3 in AUPRC). I feel a little confused about it as the baseline method also uses the GRN dataset information. Is there any reason for it?

There are some minor suggestions for writing:
1. The meaning of several words in Section 4 is not mentioned. E.g., what is the Origin method in Table 1(I guess it means using scFM without a projection function). What is NSP, L/G, SPC NET?

2. The word "Ablation study" is not suitable for Table 3-5. Maybe use "Sensitivity study".

**Strengths And Weaknesses:**

The paper provides a framework to improve existing scFM to perform GRN inference by introducing an additional nonlinear function to project the previous "in silico perturbation" results to the GRN prediction results. The paper is well writtern and the idea is interesting, although straightforward.  The authors also made many experiments to prove the results for many "unseen" cases.  I only have one major concern:

The results of the proposed GDT method provide a huge improvement over the baseline method in STR network (as much as 0.2-0.3 in AUPRC). I feel a little confused about it as the baseline method also uses the GRN dataset information. Is there any reason for it?

There are some minor suggestions for writing:
1. The meaning of several words in Section 4 is not mentioned. E.g., what is the Origin method in Table 1(I guess it means using scFM without a projection function). What is NSP, L/G, SPC NET?

2. The word "Ablation study" is not suitable for Table 3-5. Maybe use "Sensitivity study".

---

> ### Author Rebuttal · Authors · 2026-03-30
>
> We thank the reviewer for recognizing the value of our Universal GRN (UGRN) framework and our proposed methods. We sincerely appreciate the constructive feedback and address all concerns below.
>
> **W1 The Improvement of GDT over Baselines.** First, we would like to reiterate that our primary contribution is the establishment of the novel UGRN setting, a highly challenging task under traditional GRN paradigms. The recent advancement of single-cell Foundation Models (scFMs) and their generalizable gene embeddings makes this transfer setting feasible. Constructing the UGRN benchmark provides two main benefits: (1) it serves as a rigorous testbed to evaluate whether scFMs capture gene-to-gene interactions, and (2) it paves the way for cross-network GRN transfer. Although our current evaluation is primarily based on the BEELINE framework, our framework is universally applicable to any GRN transfer scenario.
>
> The performance gap between our methods and the baselines can be explained by how each method extracts information. Given that the input (training/testing sets) and the frozen scFM are identical across all methods, the difference lies entirely in how effectively each method elicits generalizable relational features from the scFM's internal representations.
>
> * **Limitations of the Baselines:** Ignoring zero-shot approaches (which are already proven ineffective for GRN Inference), our Baseline-Pert mainly relies on the traditional perturbation, i.e., setting a gene to zero and measuring the expression change in another gene. This implementation has two critical flaws. First, it generates only a single, point-estimate feature, making it highly susceptible to noise. Second, because natural gene expression values are often inherently low, the delta induced by zero-masking is marginal and easily drowned out by noise, leading to poor feature quality. Our Baseline-Emb uses static vocabulary gene embeddings, while it establishes a strong baseline, it still fails to extract dynamic cross-gene relationships induced from scFMs.
> * **The Advantage of GDT:** We first introduced Virtual Value Perturbation as a stronger variant of traditional perturbation, addressing the limitations of Baseline-Pert by applying multiple virtual perturbations with distinct, large initial values. Building upon this, GDT serves as an even more robust feature descriptor that extracts a deeper representation of gene interactions. While VVP essentially captures the average gradient (mean slope) of the perturbation response, GDT explicitly extracts the instantaneous gradient between genes modeled within the scFM. This dynamic, continuous trajectory makes GDT significantly more sensitive and generalizable for describing complex regulatory relationships, driving its superior performance in our UGRN setting.
>
> **W2 Missing Definitions and Terminology Change.** We apologize for omitting these definitions and will incorporate them clearly into the revised manuscript.
> * **Origin:** Refers to the original zero-shot predictions derived directly from the scFM (using raw attention weights or single-point perturbation values) without training the projector on the training data.
> * **STR, NSP, SPC, L/G:** These denote the four distinct biological sources. Following the BEELINE framework, these abbreviations are STRING networks, Non-SPecific ChIP-seq networks, SPeCific ChIP-seq networks, and Loss-of-Function/Gain-of-Function perturbation networks, respectively.
> * **Terminology Update:** We agree with the suggestion. "Sensitivity study" is a much more appropriate descriptor for evaluating model robustness across species, data sources, and networks. We will update the terminology for Tables 3, 4, and 5 in the revised version.

---

> > ### Author Rebuttal · Reviewer_Xch5 · 2026-03-31
> >
> > Thanks for your response. I decide to keep the score.

---

### Official Review · Reviewer_ETxE · 2026-03-07

**Soundness:** 3
**Presentation:** 4
**Significance:** 3
**Originality:** 3
**Overall Recommendation:** 5
**Confidence:** 4

**Summary:**

The contribution of this paper is two-fold: (1) the authors show that single-cell foundation models (scFMs) possess rich information that can be exploited to reconstruct gene regulatory networks (GRNs), and (2) they propose a framework to reconstruct GRNs in a supervised way, by training a model ("translator", in the form of a MLP) that maps pair-wise features $e_{ij}$ (for a gene pair $g_i$/$g_j$) derived from a (frozen) scFM to a probability of interaction between the two genes. To train such models, interaction labels are hence needed for different gene pairs.

Within the proposed framework, two methods are proposed to obtain the pair-wise features:
- Virtual Value Perturbation (VVP), where the features $e_{ij}$ are the changes of gene $g_j$ in the scFM output when changing the value of gene $g_i$ from some base virtual value to multiple other virtual values.
- Gradient Trajectory (GDT), where the features $e_{ij}$ are the gradients of the scFM output along a series of virtual values.

Eventually, they propose an ensemble method that averages the MLP prediction obtained from the two types of pair-wise features.

Multiple experiments were performed with the BEELINE framework (which provides simulated human and mouse single-cell data from synthetic networks), showing the high potential of exploiting scFMs and showing that their proposed methods outperform relevant baselines.

**Compliance With Llm Reviewing Policy:**

Affirmed.

**Final Justification:**

The rebuttal addressed all the concerns I had. I hence raised my score accordingly.

**Key Questions For Authors:**

Related to the strengths and weaknesses discussed above, my key questions are:

1. Could you discuss the computational cost of your approach (including the inference time for the scFM with respect to the MLP training)? This would probably positively highlight the efficiency of the method.

2. How do you explain the lower performance observed in the cross-network study (Table 5)?

3. Could you provide the missing details: vocabulary sizes, proportions of genes shared between the training and test sets, and the number of gene pairs in the different training and test sets?

4. In section 4.2, could you clarify what you mean by "incorporate information from both directions when constructing the pair-wise feature $\mathbf{e}_{ij}$"? For example, for VVP, does it mean that you compute the change of $g_j$ when changing $g_i$, and then compute the change of $g_i$ when changing $g_j$?

**Limitations:**

Yes.

**Strengths And Weaknesses:**

Strengths:

- I believe one of the major strengths of the proposed method is that it no longer requires expression data to reconstruct the GRN. Although some prior information is still needed (i.e., whether a subset of gene pairs interact or not), the ability to move away from expression matrices is a substantial advantage and marks a clear departure from previous GRN inference approaches, which were mostly expression-based. Some previous works have attempted to follow this direction, but the method presented in this paper demonstrates superior performance. I hence think that this paper will have a very good impact in the computational biology community.

- The experiments with the BEELINE framework are well-designed and the chosen baselines are relevant. The authors also performed multiple ablation studies, examining the generalization capabilities of their method on distinct data sources, species and networks. All this makes the empirical evaluation convincing and provides a clear picture of the advantages of the proposed method.

- Although the authors do not explicitly discuss the computational cost of their approach, it seems to be very lightweight in practice, requiring only to train two small MLPs (for the ensemble approach) on top of a frozen scFM.

- Globally, the paper is well-structured and clearly written. It was a very pleasant read.


Weaknesses:

-  The experiments are conducted only on simulated data from synthetic GRNs. While obtaining real ground-truth GRNs (to which one can compares GRN predictions) is undeniably challenging, it remains important to evaluate the method in more realistic biological settings. This is particularly relevant given the results in Table 6, which show a clear degradation of performance under more realistic conditions where the positive and negative classes are highly imbalanced.

- The discussion of the cross-network results of Table 5 is rather limited. While the proposed method generalizes well across datasets, and every particularly well across human and mouse, the cross-networks results are more mixed. This may suggest that the MLP primarily capture network-specific characteristics rather than truly general regulatory mechanisms. The authors should discuss this point explicitly and discuss its implications for the claimed universality of the approach.

* They authors mention that the approach is limited by the size of the scFM vocabulary. However, it is not mentioned to which extent the vocabularies of the two scFMs used in the experiments cover the gene sets of the different datasets.

* The paper does not report the proportion of genes that are shared between the training and test sets (which would obviously be relevant only when training and test sets are from the same species). Although the authors specify that gene pairs do not overlap between training and test sets, nothing is said about overlap at the level of individual genes. As a result, it is impossible to determine whether the reported generalization comes from unseen pairs among previously seen genes or from pairs involving completely unseen genes.

* Also, the sizes of the various training and test sets are not provided. Hence one cannot assess how much supervision is required to achieve the observed level of generalization.

---

> ### Author Rebuttal · Authors · 2026-03-30
>
> We thank the reviewer for recognizing the utility of our proposed UGRN setting, the effectiveness of our methods, specifically pointing out our methodology will have a good impact on the computational biology community. We will address all concerns below.
>
> **W1 Clarification on Data.** We would like to clarify that our evaluation strictly uses genuine in vivo and in vitro experimental data, not simulated data. We specifically utilized the 'Experimental' scRNA-seq track of the BEELINE framework (GEO accessions in Appendix B.1). Furthermore, all ground-truth reference networks are rigorously curated from authentic biological databases and physical experiments, entirely avoiding synthetic networks.
>
> Regarding the performance under imbalance, we agree that real-world GRNs are highly sparse. As shown in Table 6, while the absolute AUPRC naturally drops under extreme imbalance (e.g., 1:10), our method yields a stable absolute performance margin of ~20% over the Baseline-PERT across all N/P ratios, justifying the effectiveness of our method.
>
>
>
> **W2 & Q2 Cross-Network Results.** The stronger performance in Table 4 (cross-species) is because the training set inherently contains all four network types, making the setting relatively easier. Table 5 presents a highly challenging cross-network task. The reason is that different ground-truth networks define "regulation" in fundamentally different ways. For example, STRING represents protein interactions or co-expression, whereas ChIP-seq represents physical DNA binding. Training a model to learn "physical binding" rules and then asking it to predict "protein interactions" introduces a massive semantic gap. Because our method consistently outperforms strong baselines by a relatively stable margin, we believe the performance fluctuations are attributable to the difficulty of different settings rather than a lack of universality in our method.
>
>
>
> **W3, W4 & Q3 Vocab and Overlap Details.**
> To ensure the standardized benchmarking of our framework, we directly applied the widely adopted BEELINE framework. We apologize for omitting the intersection data, as we noted that the gene interactions within these datasets are highly sparse.
>
> The scFMs we applied follow scGPT and scbenchmark, sharing the same vocabulary with a size of 60697. This vocabulary covers 89.97% of the gene sets across the different datasets (on average). The proportion of genes shared between the training and test sets is 9.51% (on average), and the number of gene pairs in the respective training and test sets could be found in the Appendix.
>
> To further address the concern about generalization, we conducted an additional experiment based on scbenchmark where we strictly eliminated all seen genes in the training when conduct testing, (other settings are identical to Table 1). The AUPRC results are as follows:
>
> | Method | Baseline-PERT | Baseline-EMB | Ours (VVP) | Ours (GDT) | Ours (Ens) |
> | :--- | :--- | :--- | :--- | :--- | :--- |
> | Table 1 | 0.596 | 0.663 | 0.663 | 0.738 | 0.768 |
> | Strict| 0.608 | 0.590 | 0.651 | 0.731 | 0.756 |
>
> While the embedding baseline is influenced by this strict setting, the remaining results further confirm that our methods' effectiveness genuinely stems from our novel approach to extracting inter-gene features from scFMs. We thank the reviewer for this constructive suggestion,  and we will add the above details and discussion to the revision.
>
>
> **W5 Training Data Size and Supervision Requirements**
> As detailed in Appendices A.1 and A.2, our method achieves consistent improvements across varying training dataset sizes. To further validate this, we trained the MLP using varying ratios of the hESC dataset (Table 1 setting):
>
> | Ratio | 10% | 20% | 40% | 60% | 80% | 100% (Table 1) |
> | :--- | :--- | :--- | :--- | :--- | :--- | :--- |
> | AUPRC | 0.736 | 0.748 | 0.757 | 0.767 | 0.768 | 0.768 |
>
> Our method maintains strong performance even with heavily reduced training data, demonstrating high data efficiency without requiring massive supervision.
>
> **Q1 Computational Cost**
> Because the scFM is relatively small and frozen, the downstream MLP training is fast, and methods like VVP and GDT act as data preprocessing steps. Therefore, their computational overhead is acceptable. To provide a concrete example based on Table 1 (for VVP): on a single NVIDIA RTX 4090 GPU, the VVP feature extraction takes approximately 6.5433 seconds average per dataset, and the subsequent MLP training takes only 7.1919 seconds (50epochs).
>
> **Q4 Clarification on "incorporating information from both directions"**
> Yes, the understanding is exactly correct. Since scFMs primarily capture gene correlations rather than direct causality, incorporating both directions allows the MLP to learn much more comprehensive representations.

---

> > ### Author Rebuttal · Reviewer_ETxE · 2026-04-01
> >
> > Thank you for your response and clarifications.
> > I raised my overall recommendation score.

---

### Official Review · Reviewer_7yPp · 2026-03-12

**Soundness:** 2
**Presentation:** 3
**Significance:** 2
**Originality:** 3
**Overall Recommendation:** 3
**Confidence:** 4

**Summary:**

This paper proposes a framework for "universal" gene regulatory network (GRN) inference using single-cell foundation models (scFMs). The authors argue that existing methods for extracting GRN predictions from scFMs are too simplistic, and propose two alternatives: 1) Virtual Value Perturbation, which probes the model at artificial inputs, and 2) Gradient Trajectory, which extracts gradients of model outputs across a range of such inputs. These produce pairwise gene features fed to an MLP classifier to predict regulatory links. The authors evaluate generalization to unseen genes and datasets using a leave-one-dataset-out protocol, benchmarking against association and binding databases treated as ground-truth regulatory networks.

**Compliance With Llm Reviewing Policy:**

Affirmed.

**Final Justification:**

I appreciate the authors' responsiveness throughout the discussion. The rebuttal clarified a more modest goal than the paper's framing around "unlocking regulatory knowledge" and "universal GRN inference" suggests — probing whether scFMs encode generalizable inter-gene representations. This is reasonable, but in tension with the paper's claims.

The proposed method probes the scFM with constant-valued vectors where every gene shares the same expression, extracting a single fixed feature per gene pair regardless of cellular context. The premise of scFMs is that they encode context-dependent regulatory logic from diverse cellular states; by construction, this probing regime cannot access any of that. The authors acknowledge in Section 4.4 that real regulation is "strictly context-dependent" but treat this as future work — yet the paper's approach implicitly assumes that the relevant knowledge takes the form of a static, context-independent representation, an assumption that is not discussed.

As discussed during the rebuttal, the use of constant-valued inputs that are far out-of-distribution for models trained on real expression profiles remains a concern. Without a comparison against biologically grounded references (e.g., VVP anchored to cell-type median expression profiles), it is unclear whether the OOD regime is necessary or beneficial.

The empirical pattern reinforces the concern. VVP achieves identical performance to Baseline-Emb (0.663 vs 0.663), which simply uses the learned gene vocabulary embeddings without any forward pass. GDT improves over baselines on associative sources (STRING, nonspecific ChIP-seq) but not on cell-type-specific ChIP-seq, suggesting the methods capture broad gene-gene associations encoded in model parameters rather than regulatory logic.

I maintain my score of 3.

**Key Questions For Authors:**

See weaknesses above. In particular, what evidence is there that the model's response to artificial, constant-valued inputs reflects regulatory biology rather than generic properties of the model's gene representations? It is notable that VVP performs comparably to or below the simple embedding baseline, raising the question of what the virtual value probing adds.

**Limitations:**

yes

**Strengths And Weaknesses:**

### Strengths

* The paper tackles a genuine problem: current methods for extracting regulatory signals from scFMs are crude and not very principled. VVP and GDT are technically interesting approaches that probe model behavior more systematically.
* The experimental evaluation is relatively thorough in its own terms, including cross-dataset, cross-species leave-one-dataset-out comparisons.
* The paper is clearly written, and Figure 2 in particular effectively communicates the methods.

### Weaknesses

* __The paper presents a very narrow view of what GRN inference is.__ The authors state that *"the fundamental objective of GRN inference is to learn a parameterized mapping function $f_\theta$ from expression vectors to regulatory probabilities"*, i.e. $f_\theta (\mathbf{x}_i, \mathbf{x}_j)$, but GRN inference in general is a much broader problem. It concerns uncovering *causal*, directed, context-dependent regulatory relationships between genes, and the modelling approaches in the field are correspondingly broader than pairwise binary classification. The "open-world" versus "closed-world" framing is also somewhat artificial: in practice, as long as the method infers edges from expression data, it can be applied to any new dataset without difficulty. The genuinely interesting question is whether regulatory knowledge transfers meaningfully across biological contexts, but the paper never interrogates what is actually transferring.
* __It is unclear whether the evaluation measures what the paper claims.__ The ground-truth sources used (STRING, ChIP-seq, LOF/GOF) are useful resources, but they are not gene regulatory networks in the causal sense. STRING is primarily an association database, ChIP-seq captures physical binding rather than functional regulation, and the one causal source (LOF/GOF) covers only a single dataset. The fact that performance is strongest on STRING (the most associative and least regulatory source) raises the question of whether the method is learning regulatory logic or broader gene-gene association patterns, particularly given that it predicts undirected edges rather than directed regulation. This concern is compounded by the use of artificial, context-free inputs (see next point).
* __VVP and GDT extract features from biologically unrealistic inputs.__ Both methods probe the scFM using artificial expression vectors where all genes are set to the same constant value — no real cell resembles this. The model was trained on real expression profiles, and its behavior on such out-of-distribution inputs is undefined. This design also explicitly discards cell-type and cell-state specificity, which has been considered one of the main potential advantages of using scFMs for GRN inference.
* __The paper does not engage with experimental perturbation data.__ The field has increasingly moved toward Perturb-seq and related screens as both a source of causal ground truth and a direct way to study gene regulation. Perturbation data could potentially also serve as a training signal for adapting scFMs to predict regulatory effects. The absence of this perspective in a universal GRN inference paper is a significant gap.

---

> ### Author Rebuttal · Authors · 2026-03-30
>
> We thank the reviewer for recognizing that our paper tackles a genuine problem, finding our VVP and  GDT methods to be technically interesting approaches for systematically probing model behavior, and appreciating our thorough cross-dataset and cross-species evaluations.
>
> We realize our naming "UGRN" may have been overly broad, leading to misunderstandings about our scope. We would like to clarify that our primary motivation is to establish a transfer setting to probe whether scFMs genuinely capture transferable inter-gene relationships. We address other concerns below.
>
> **W1 Narrow view of GRN inference.** We respectfully disagree that the open-world challenge is artificial or that inferring edges on new datasets is straightforward. As shown in Figure 1, the key difficulty for traditional GRN methods in an open-world setting is the inconsistency in expression vector lengths (differing gene sets) across datasets. This structural mismatch prevents models trained on one dataset from being directly applied to another, which is why prior methods are typically trained and tested within the same dataset (such as scGREAT and InfoSEM).
> By proposing the UGRN framework, we decouple interaction prediction from dataset-specific expression matrices, allowing the model to act as a universal plug-in. Our motivation is not to transfer specific biological mechanisms across contexts, it is to find the generalizable inter-gene representations of scFMs which could capture the regulatory knowledge between genes.
>
>
> **W2 Undirected edges and Ground Truth Evaluation.** We would like to clarify a critical misunderstanding: our model explicitly predicts directed regulatory relationships, not undirected associative edges. Our evaluation specifically focuses on interactions outgoing from transcription factors (TFs). Regarding ground truth, we deliberately utilized four diverse sources to prevent the model from merely memorizing co-expression patterns. While STRING includes associative data, our LOF/GOF benchmark is explicitly based on experimentally validated, perturbation-derived causal interactions. The robust performance of our methods on this causal LOF/GOF data demonstrates that the model captures intrinsic regulatory logic rather than just broad association patterns. Finally, variations in absolute performance across sources reflect the inherent difficulty of the respective settings, not a failure of the model. Our methods consistently and stably improve upon baselines across all evaluated settings.
>
> **W3 & Q1 the concern about VVP.** The use of counterfactual, constant-valued inputs is a deliberate design choice rather than a limitation. Traditional biological in-silico perturbations perform poorly on scFMs (as shown in Table 1&2). Constant inputs provide a controlled environment to probe the model's internal knowledge, isolating inter-gene relationships from dataset-specific expression variations. As Reviewer naUQ insightfully noted, this uniform setting is "a clever way to reduce dataset-specific scale mismatch." By breaking away from traditional biological input constraints, we adopt a probing strategy similar to those used in LLM research, successfully unlocking the scFM's latent capacity to represent gene relationships.
>
> **W4 Absence of Perturb-seq.** We apologize for the confusion caused by our broad UGRN naming. Our immediate goal was not to establish a singular, universally applicable GRN for all biological contexts, but to create a robust transfer setting to validate scFMs. We agree that large-scale perturbation screens are the future standard for causal GRN inference. However, the field currently lacks a unified and standardized benchmark for these novel datasets that allows for comprehensive algorithmic evaluation. Therefore, we evaluated our method using BEELINE, a rigorously defined and widely adopted benchmark utilized by contemporary GRN studies (e.g., scGREAT and InfoSEM). Importantly, our UGRN framework and proposed methods are highly adaptable and can be seamlessly extended to perturbation-based downstream tasks as standardized benchmarks become available.
>
> **Q2 VVP performance compared to the embedding baseline.** It is important to evaluate VVP against its direct counterpart, which is the traditional biological perturbation method (Baseline-Pert). Against this standard, VVP shows comprehensive and stable improvements, proving its necessity as an advanced in-silico perturbation technique for feature extraction of scFMs. Compared to our proposed strong baseline embedding, VVP still achieves SOTA performance on some settings (e.g., Spc in Table 1).

---

> > ### Author Rebuttal · Reviewer_7yPp · 2026-04-02
> >
> > Thank you for the detailed response. I appreciate the authors' clarifications, but several core concerns remain unresolved. The authors clarify their goal as probing whether scFMs capture transferable inter-gene representations, rather than transferring regulatory mechanisms. This is reasonable, but sits in tension with the paper's claims about unlocking regulatory knowledge and GRN inference.
> >
> > **On virtual inputs.** I understand this is a deliberate design choice, but that does not address the concern. The scFM was trained on real expression profiles; its behavior on constant-valued vectors where all genes share the same value is OOD and therefore undefined. The analogy to LLM probing does not resolve this — LLM probing uses real, in-distribution text, not artificial inputs where every token is identical. There is no reason to expect the model's outputs in this regime to be meaningful. This remains my most fundamental concern with the paper.
> >
> > **On directed edges.** The authors state they "explicitly predict directed regulatory relationships." However, as confirmed in response to Reviewer ETxE (Q4), features are computed in both directions and concatenated, and the model outputs a single score per pair. The model itself does not distinguish direction; it is imposed by the evaluation protocol (restricting to TF-outgoing pairs), not learned.
> >
> > **On Perturb-seq.** I appreciate the acknowledgment that perturbation screens are the future standard. However, Perturb-seq datasets have been available for some time, and there are existing efforts to derive GRN ground truths from them (e.g., CausalBench by Chevalley et al 2022). I would expect a paper on universal GRN inference to engage with this data, whether as ground truth, as a point of comparison, or as a training signal for adapting scFMs.

---

> > > ### Author Response · Authors · 2026-04-03
> > >
> > > Thank you for your follow-up questions. We would like to clarify our primary objective: to investigate whether and how scFMs contain generalizable inter-gene representations. To test this, we proposed the UGRN framework and evaluated it using the BEELINE. The definition of the gene regulatory relationships in our evaluation is directly inherited from BEELINE datasets, a widely adopted standard in the field [1-6]. While we appreciate the suggestion on the strict definition of GRNs, re-evaluating this biological standard is beyond the scope of our study.
> > >
> > > **1. Virtual Inputs.** (a) Foundation Model Capabilities. Like large foundation models in other domains, scFMs are pre-trained at scale, enabling them to encode meaningful information even when presented with out-of-distribution (OOD) or counterfactual inputs.
> > > (b) Feature Extraction vs. Direct Output. We do not directly use the model's raw output from these virtual inputs (which we agree would lack meaning). Instead, we use them as a controlled stimulus to extract internal features that might be hidden under standard biological inputs. This is a widely used probing technique. For example, in Large Vision-Language Models (LVLMs), inputs like pure noise or blank images are frequently used to probe internal model behaviors [7,8].
> > > (c) The LLM Analogy. Regarding the analogy to LLMs, comparing our virtual values to a sequence of "identical tokens" is a slight mismatch. The more accurate analogy is at the sample (sentence) level rather than the token level. Each virtual gene value is a seen scalar (like a known token), but their combination forms a counterfactual or OOD expression profile (like an ungrammatical or counterfactual sentence used to probe the LLM [9]).
> > >
> > > **2. Directed Edges.** We apologize for the ambiguity. We agree that the model's raw internal predictions are not inherently causal or directed. In fact, our bidirectional implementation (computing features in both directions) was with the understanding that scFMs lack inherent directionality. The "directed regulatory relationships" are the result of the downstream MLP mapping and the specific evaluation protocol, not a direct output of the scFM itself. We will revise this in the revision to avoid misleading.
> > >
> > > **3. Perturb-seq Data.** We appreciate your reference to CausalBench and agree that engaging with large-scale perturbation screens is a vital trajectory.
> > > (a) There is a fundamental divergence in evaluation objectives. Perturb-seq benchmarks (e.g., CausalBench) are predominantly designed for predicting continuous post-perturbation expression profiles or inferring causal graphs strictly confined to a small subset of directly perturbed genes. In contrast, GRN datasets (e.g. BEELINE) aim to perform systemic, genome-wide Transcription Factor (TF)-to-target inference. Assessing transcriptome-scale regulatory mappings requires a benchmark explicitly formulated for global edge prediction, which CausalBench is not currently optimized to provide.
> > > (b) Our proposed method and framework are dataset-agnostic, seamlessly applicable to various datasets. We utilized BEELINE because it is standardized and widely adopted [1-6]. We will clarify in the revision to highlight that our framework has the broad applicability across downstream GRN inference datasets, rather than claiming to resolve all forms of biological causality between genes.
> > >
> > > ## References
> > > 1. Wang, Yuchen, et al. "scGREAT: Transformer-based deep-language model for gene regulatory network inference from single-cell transcriptomics." Iscience 27.4 (2024).
> > > 2. Chen, Guangyi, and Zhi-Ping Liu. "Graph attention network for link prediction of gene regulations from single-cell RNA-sequencing data." Bioinformatics 38.19 (2022): 4522-4529.
> > > 3. Mao, Guo, et al. "Predicting gene regulatory links from single-cell RNA-seq data using graph neural networks." Briefings in Bioinformatics 24.6 (2023): bbad414.
> > > 4. Kommu, Sindhura, et al. "Prediction of gene regulatory connections with joint single-cell foundation models and graph-based learning." Bioinformatics 41.Supplement_1 (2025): i619-i627.
> > > 5. Yu, Weiming, et al. "GCLink: a graph contrastive link prediction framework for gene regulatory network inference." Bioinformatics 41.3 (2025): btaf074.
> > > 6. Wang, Kai, et al. "GRLGRN: graph representation-based learning to infer gene regulatory networks from single-cell RNA-seq data." BMC bioinformatics 26.1 (2025): 108.
> > > 7. Leng, Sicong, et al. "Mitigating object hallucinations in large vision-language models through visual contrastive decoding." Proceedings of the IEEE/CVF Conference on Computer Vision and Pattern Recognition. 2024.
> > > 8. Suo, Wei, et al. "Octopus: Alleviating hallucination via dynamic contrastive decoding." Proceedings of the Computer Vision and Pattern Recognition Conference. 2025.
> > > 9. Cherepanova, Valeriia, and James Zou. "Talking Nonsense: Probing Large Language Models' Understanding of Adversarial Gibberish Inputs." arXiv preprint arXiv:2404.17120 (2024).

---

### Official Review · Reviewer_naUQ · 2026-03-13

**Soundness:** 2
**Presentation:** 3
**Significance:** 3
**Originality:** 2
**Overall Recommendation:** 4
**Confidence:** 3

**Summary:**

This paper studies whether frozen single-cell foundation models (scFMs) can support gene regulatory network (GRN) inference beyond the usual setting. The paper argues that standard scFM readouts such as single-point perturbation or attention are poorly aligned with GRN prediction, and proposes a new Universal GRN (UGRN) setting that evaluates cross-dataset generalization.

Methodologically, the paper introduces two feature engineering modules from frozen scFMs: Virtual Value Perturbation (VVP), which probes responses across multiple virtual source-gene values, and Gradient Trajectory (GDT), which aggregates gradient-based sensitivities via model logits. These features are then mapped to regulatory probabilities by a small MLP projector.

Experiments on seven RNA-seq datasets with leave-one/some-dataset-out, cross-species, and cross-network evaluations show improved AUPRC over perturbation, attention, and embedding baselines, using both scGPT and scBenchmark as backbones.

**Compliance With Llm Reviewing Policy:**

Affirmed.

**Final Justification:**

The author's rebuttal addressed all my concerns, especially for the train/test split overlap. It seems that they follow the standard benchmark, which does not have much overlap between train/test. Also, they test on a new benchmark without data overlap and their method performs well. Their method is simple yet effective and can serve as a strong baseline in the future for this important GRN prediction problem.

**Key Questions For Authors:**

1. Could the authors report runtime/memory and, ideally, a compute-matched comparison?

2. Could the authors better justify the reliability of the chosen GRN “ground truths”?

3. Can the authors evaluate a stricter zero-shot setting, or at least discuss more clearly why the supervised MLP-projector setting is the right target use case? As it stands, the paper seems to show that VVP/GDT provide better features for supervised alignment, not necessarily that frozen scFMs alone can infer GRNs when labels are unavailable.

4. Please clarify the biological mechanism behind cross-species transfer for the human and mouse species. Are genes aligned via orthologs, shared tokens, or some other vocabulary design?

**Limitations:**

(1) the gap between the supervised projector setting and strict zero-shot deployment, and (2) the biological assumptions underlying cross-species transfer. See details in Weakness.

**Strengths And Weaknesses:**

**Strengths**

Generally, I think the paper has clear strengths.
- The problem is important, the empirical setup is broader than a standard within-dataset split, and the paper evaluates two scFM backbones with several ablations.
- I also find VVP genuinely interesting: using a shared virtual reference state to probe inter-gene effects is a clever way to reduce dataset-specific scale mismatch.
- The cross-dataset/cross-network/cross-species evaluations are also helpful and make the paper more ambitious than a standard supervised GRN paper.

**Weakness.**

- The performance gain seems at least partly tied to higher query-time compute. VVP performs multiple model evaluations per gene pair, and GDT aggregates gradients along a trajectory. In the default continuous-input setting, VVP uses 10 virtual target values and GDT uses a step size of 0.1, i.e., substantially more probing than single-point perturbation baselines. I would therefore like to see either runtime/memory comparisons or a compute-matched baseline. As written, part of the improvement may reflect “more inference budget for probing” rather than a purely better feature design.

- The paper relies on imperfect GRN ground truth. This is not unique to this submission, but it matters for the strength of the paper’s claims. BEELINE is indeed a widely used benchmark in recent GRN papers, but its labels are proxies built from cell-type-specific ChIP-seq, non-specific TF–target resources, and STRING. More broadly, benchmark papers such as GRNbenchmark and geneRNIB also emphasize that GRN benchmarking lack a single gold standard and that context specificity is a central challenge. I therefore think the paper should work harder to convince readers that the selected “ground truths” are reliable enough, or at least state more clearly what kind of regulatory signal is actually being evaluated.

- I am not convinced by the problem setting relative to realistic deployment. The most natural application of scFMs is often a strict zero-shot or label-scarce setting: pretrain on large single-cell corpora, then directly transfer to new tasks or datasets. However, the main results depend on supervised training of an MLP projector using GRN labels. That means, the current experiments answer the question “which derived features are easier for an MLP to align with existing GRN labels?”. And the UGRN framework seems not able to work on this zero-shot setting. But it does not align with real-world applications, where we don’t have GRN labels for supervised training.

- I find the cross-species setting under-justified biologically. The paper mixes human and mouse datasets under Leave-Some-Dataset-Out and also reports cross-species results. This is potentially interesting, and cross-species transfer is certainly plausible because some regulatory information is conserved. However, regulatory programs are also known to be different across species. Therefore, the dataset choce is strange: RNA-seq datasets include both human and mouse datasets. And the model is trained by “Leave-Some-Dataset-Out”, how does this transfer make sense?

- For novelty, I view VVP as the main original contribution and, in my opinion, the most interesting methodological component of the paper. By contrast, GDT feels closer to an adaptation of standard gradient-based attribution / path-based interpretation ideas to the GRN setting than a fundamentally new idea.

---

> ### Author Rebuttal · Authors · 2026-03-30
>
> We thank the reviewer for recognizing the value of our proposed framework, VVP method is genuinely interesting for addressing dataset-specific scale mismatch and appreciating the ambition of our comprehensive cross-dataset, cross-network, and cross-species evaluations.
>
> We would like to first clarify our motivation. While scFMs have advanced rapidly, their direct zero-shot performance on GRN inference remains poor. To investigate whether scFMs truly capture underlying gene regulatory relationships, we proposed the UGRN framework. Then, we further establish strong baselines (i.e., emb), designing specific methods (VVP and GDT), and demonstrating their effectiveness through comprehensive evaluations. We address all concerns below.
>
> **W1 & Q1 Concern about consumption.** Note that our feature extraction is a one-time preprocessing step, making the time cost highly manageable. VVP takes $k$ forward passes ($k$ = perturbation steps) vs. Baseline-Pert's 1 pass (0.0115s/step, batch 16, 512 genes, RTX 4090). Both share the same memory (about 2.4GB). GDT requires one additional backward pass per step (about 2.5GB, 0.0198s), maintaining a comparable memory overhead.
>
> Regarding a compute-matched baseline, we respectfully argue that it is mathematically uninformative. Traditional in-silico perturbation (Baseline-Pert) is constrained to a single manipulation (setting expression to 0). Repeating this zero-masking operation $k$ times does not yield any additional information. Our core innovation breaks this traditional path dependency by introducing virtual values.
>
> Furthermore, the performance gain is not merely a result of "more inference budget." As shown in Appendix Tables A6 and A9, the number of extraction steps has a marginal impact on the final results once a threshold is reached. This confirms that our methods succeed by effectively inducing latent relational features from the scFM, rather than relying on brute-force computation.
>
> **W2 & Q2 GRN "Ground Truths".** We agree that a perfect GRN gold standard does not currently exist. However, our primary goal is not to search for a universal ground truth, but rather to establish UGRN as a rigorous setting to detect whether scFMs encode gene interaction information. We use BEELINE, a widely adopted framework, to demonstrate the viability of our framework and to verify that scFMs do contain generalizable regulatory signals.
> The strength of our framework lies precisely in its flexibility, where it does not rigidly depend on a specific ground truth. If the biological community defines a more perfect GRN standard in the future, our framework and method can seamlessly utilize it to transfer to predict relationships between arbitrary unseen genes.
>
> **W3 & Q3 Zero-Shot Setting.** We would like to clarify that, in the UGRN setting, zero-shot refers to predicting relationships between unseen genes during inference. Here are the reasons:
> * A strict zero-shot setting (directly using frozen scFM representations) yields poor performance, as demonstrated by the "Origin" results in Table 1. While this fits the traditional definition, it has little practical value.
> * In realistic deployment, researchers often possess partial biological priors (e.g., known pathways, literature). UGRN allows users to train a lightweight MLP on this limited prior knowledge to align the inter-gene representations, which then generalizes to unseen genes.
>
> To further prove UGRN's viability in label-scarce settings, we conducted an experiment using only 10% of the training data from Table 1. Our method Ens achieved 0.736 compared to the full training as 0.768 in Table 1, confirming its practical value.
>
> **W4 & Q4 Mechanism of Cross-Species Transfer.**
> * Technically: Our pipeline leverages the unified gene vocabulary of scFMs (e.g., scGPT) by mapping mouse genes to their human orthologs via standard databases (e.g., Ensembl). Therefore, homologous genes share the exact same token IDs and latent representations, providing a unified feature space.
> * Biologically: While peripheral regulations vary, core gene regulatory networks (e.g., developmental pathways governed by highly conserved TFs like SOX2 or OCT4) exhibit strong evolutionary conservation across mammals. The premise of UGRN is not to claim identical networks between humans and mice, but rather to demonstrate that scFMs have learned a "universal regulatory feature". The high cross-species AUPRC achieved by our features (Table 4) empirically validates that this conserved logic is successfully extracted.
>
> **W5 Originality of GDT.**
> Our proposed VVP and GDT break traditional biological constraints by introducing counterfactual virtual values, capturing the interactions between genes in different level of the scFMs. GDT is not simply an off-the-shelf application of path-based attribution, but a highly tailored adaptation designed to capture complex biological dynamics.

---

> > ### Author Rebuttal · Reviewer_naUQ · 2026-04-02
> >
> > (New) Thanks for the further answering. My questions are all well resolved. Thanks for the interesting work. I decide to raise my score to 4.
> >
> > (Original)Thanks for the efforts and response. About the zero-shot setting, or more specifically, your train/test setting, do you have any statistics to show that **the label overlap between datasets and the train/test split**? Normally, MLP + supervised learning performs good because the test setting is too easy. Other baselines are all "inference-only" methods, and supervised learning outperforms them is not so surprising. In other words, show that the train/test split is really meaningful, where the training labels should not show a lot of information for the test labels.

---

> > > ### Author Response · Authors · 2026-04-02
> > >
> > > Thank you for the constructive follow-up. We address all concerns below.
> > >
> > > 1. Clarification on Baselines.
> > > The "inference-only" results in our paper merely serve to show that scFMs cannot perform GRN inference well. Our primary comparisons are against our proposed baselines (e.g., Baseline-Pert and Baseline-Emb).
> > >
> > > 2. Overlap Statistics.
> > > To ensure a standardized evaluation, we directly applied the widely adopted BEELINE framework. We apologize for omitting the overlap statistics in the manuscript. The average label overlap between the train and test splits is only 1.12%. Additionally, as addressed regarding Reviewer ETxE (W3, W4 & Q3), the average gene overlap is only 9.51%.
> > >
> > > To further address the concern, we conducted an additional "Strict" experiment on scBenchmark, strictly eliminating all seen labels/genes in the training when conducting testing:
> > >
> > > | Method | Baseline-Pert | Baseline-Emb | Ours (VVP) | Ours (GDT) | Ours (Ens) |
> > > | :--- | :---: | :---: | :---: | :---: | :---: |
> > > | Table 1 | 0.596 | 0.663 | 0.663 | 0.738 | 0.768 |
> > > | Strict-label | 0.591 | 0.645 | 0.658 | 0.735 | 0.767 |
> > > | Strict-gene| 0.608 | 0.590 | 0.651 | 0.731 | 0.756 |
> > >
> > > We observe that eliminating the overlap labels has a limited impact on the overall performance. Furthermore, while the Baseline-Emb is influenced by this strict setting (eliminating seen genes), the remaining robust results prove that our methods do not simply overfit to seen genes or labels. Instead, they successfully induce transferable inter-gene features from the scFM, enabling superior GRN inference on entirely unseen genes. We thank the reviewer for this constructive suggestion, and we will add the above details and discussion to the revision.

---

### Decision · Program_Chairs · 2026-04-30

**Decision:**

Accept (regular)

**Comment:**

This paper makes two complementary contributions to the problem of Gene Regulatory Network (GRN) inference using single-cell foundation models (scFMs). First, it introduces a Universal GRN (UGRN) benchmark that evaluates regulatory predictions under a cross-dataset generalization setting: predicting relationships between gene pairs not seen during training, and across datasets, species, and network types.
This is a more stringent and realistic evaluation than the standard within-dataset protocol used by most prior work.
Second, it proposes two novel feature distillation strategies, Virtual Value Perturbation (VVP) and Gradient Trajectory (GDT), which probe frozen scFMs using counterfactual, constant-valued input vectors to extract generalizable inter-gene relational features. These features are then mapped to regulatory probabilities by a lightweight MLP. Extensive experiments across seven RNA-seq datasets demonstrate that the ensemble approach substantially outperforms perturbation and embedding baselines.

Assessment:
After reviewing the full discussion, I find that the weight of evidence supports acceptance. The UGRN benchmark addresses a genuine gap: cross-dataset generalization in GRN inference, where varying gene vocabularies between datasets make direct model transfer impossible, is an underappreciated and practically important challenge. The proposed VVP/GDT methods are simple yet effective and, as Reviewer ETxE notes, represent a meaningful departure from expression-dependent GRN inference. The cross-dataset, cross-network, and cross-species evaluation is more comprehensive than most published GRN papers, and the new strict-gene experiment in the rebuttal significantly reduces concerns about overfitting to seen genes.
I take Reviewer 7yPp's concern about OOD inputs seriously. It is true that the scFM was trained on real expression profiles and its behavior under constant-valued inputs is not guaranteed to be well-characterized. However, the empirical evidence shows consistent improvement over baselines even in the strict-gene and strict-label settings, and the analogy to LLM probing with out-of-distribution inputs is reasonable. The claim that GDT captures "regulatory knowledge" should be appropriately moderated in the final manuscript, and the authors indicated they will revise framing accordingly.
The one remaining issue to flag for the camera-ready is the claim about "directed regulatory relationships." As clarified during the rebuttal, directionality is imposed by the evaluation protocol (TF-outgoing restriction) rather than learned by the model, and the manuscript should be revised to make this explicit.

Recommended Revisions for Camera-Ready

1. Revise the framing of the main contribution from "unlocking regulatory knowledge" to the more accurate "probing whether scFMs encode generalizable inter-gene representations for GRN inference."
2. Clarify that the method predicts relationships restricted to TF-outgoing pairs by the evaluation protocol, not that the model learns inherently directed edges.
3. Include the train/test gene and label overlap statistics (9.51% gene overlap, 1.12% label overlap) and the strict-gene/strict-label ablation results in the main paper or appendix.
4. Add missing definitions for abbreviations (NSP, SPC, L/G, Origin) and relabel "Ablation study" in Tables 3–5 as "Sensitivity study" as suggested by Reviewer Xch5.